# The suprachiasmatic nucleus regulates brown fat thermogenesis in male mice through an adrenergic receptor ADRB3-S100B signaling pathway

Yizhun Zeng[1], Xiaopeng Song[1], Qi Chen[1], Yue Gu[1], Jie Zhang[1], Tao Zhou[1], Zhihao Li[1], Tao Wang[1], Le Chang[1], Hongwei Yao[2], Yan Wang[3], Liyan Miao[3], Liujia Qian[4], Tiannan Guo[4], Yong Zhang[1], Sonia Rodriguez-Fernandez[5], Antonio Vidal-Puig[5,6]*, Ying Xu [1,7,8]*

**1** Cambridge-Su Genomic Resource Center, The Fourth Affiliated Hospital of Soochow University, Medical School of Soochow University, Suzhou, Jiangsu, China, **2** Institute of Molecular Enzymology, School of Life Sciences, Medical School of Soochow University, Suzhou, Jiangsu, China, **3** Department of Clinical Pharmacology, First Affiliated Hospital of Soochow University, Suzhou, Jiangsu, China, **4** School of Medicine, Westlake University, Hangzhou, Zhejiang Province, China, **5** University of Cambridge Metabolic Research Laboratories, Institute of Metabolic Science, MDU MRC, Cambridge, United Kingdom, **6** Centro de Investigación Príncipe Felipe, CIBERDEM, Valencia, Spain, **7** The Medical Center, The Fourth Affiliated Hospital of Soochow University, Suzhou, Jiangsu, China, **8** Jiangsu Key Laboratory of Neuropsychiatric Diseases, Medical School of Soochow University, Suzhou, Jiangsu, China

* yingxu@suda.edu.cn (YX); ajv22@medschl.cam.ac.uk (AV-P)

## Abstract

The suprachiasmatic nucleus (SCN), the central circadian pacemaker, orchestrates daily metabolic rhythms, yet its role in substrate selection and thermogenic adaptation under stress remains insufficiently understood. Here, we show that SCN lesioning abolishes the adaptive suppression of brown adipose tissue (BAT) thermogenesis typically observed during time-restricted feeding in subthermoneutral environments (TRF-STE), a paradigm that imposes concurrent nutrient and thermal stress. Contrary to wild-type responses, SCN-lesioned mice maintain elevated BAT thermogenic activity, despite impaired lipolysis, instead shifting toward glucose-driven heat production. This phenotype is accompanied by sustained sympathetic tone and β3-adrenergic receptor (ADRB3) signaling in BAT. Mechanistically, we identify a SCN-regulated ADRB3-S100B signaling axis underlying this metabolic reprogramming. S100B, a nutrient-sensitive calcium-binding protein, is upregulated in BAT following SCN disruption, where it promotes thermogenesis by stimulating brown adipocyte proliferation and suppressing senescence. Functional studies reveal that S100B is both necessary and sufficient for sustaining BAT thermogenesis under TRF-STE. Furthermore, diverse SCN disruption models, including light-induced circadian arrhythmia, N-Methyl-D-aspartic acid (NMDA) excitotoxicity, and Caspase-3-mediated ablation, consistently elevate S100B expression in BAT, reinforcing its role as a convergent effector of SCN-regulated metabolic adaptation. Thus, in intact animal, the

**Data availability statement:** We have submitted the RNA-seq data to the NCBI GEO database, accessible at https://www.ncbi.nlm.nih.gov/geo/query/acc.cgi?acc=GSE300640. SnRNA-seq data have been deposited in the NCBI BioProject database under accession PRJNA1291990. All other related data are available in the Supporting information files, namely S1 Raw Images and S1 Source Data.

**Funding:** This work was supported by National Key R&D program of China (2024YFA1803201) to YX, the National Natural Science Foundation of China (82341239, 82371487) to YX, National Key R&D program of China (2022YFA1604504) to TW, Interdisciplinary Basic Frontier Innovation Program of Suzhou Medical College of Soochow University (YXY2303022) to YX, Lingang Laboratory & National Key Laboratory of Human Factors Engineering Joint Grant (LG-TKN-202203-01) to YX, Priority Academic Program Development of the Jiangsu Higher Education Institutes (PAPD) and the National Center for International Research (2017B01012) to YX. The funders had no role in study design, data collection and analysis, decision to publish, or preparation of the manuscript.

**Competing interests:** The authors have declared that no competing interests exist.

**Abbreviations:** 6-OHDA, 6-hydroxydopamine; $^{18}$F-FDG, $^{18}$F-fluorodeoxyglucose; AAV, adeno-associated virus; ADRB3, β3-adrenergic receptor; ANCOVA, analysis of covariance; ATP, adenosine triphosphate; BAT, brown adipose tissue; CBT, Core body temperature; coIP, co-immunoprecipitation; ECAR, extracellular acidification rate; EdU, 5-ethynyl-2′-deoxyuridine; EE, energy expenditure; ERK, extracellular signal-regulated kinase; eWAT, epididymal WAT; GLM, General Linear Model; HFD, high-fat diet; HSL, hormone-sensitive lipase; IPA, Ingenuity Pathway Analysis; mtDNA, mitochondrial DNA; NE, norepinephrine; NEFA, non-esterified fatty acids; NMDA, N-Methyl-D-aspartic acid; OCR, oxygen consumption rate; OXPHOS, oxidative phosphorylation complex proteins; PET/CT, positron emission tomography/computed tomography; RER, respiratory exchange ratio; SCN, suprachiasmatic nucleus; SCNx, SCN-lesioned; snRNA-seq, single-cell nuclear RNA sequencing; SNS, sympathetic nervous system;

SCN restrains the ADRB3-S100B module, gating BAT thermogenic output in accordance with energetic availability. Disruption of SCN output lifts this restraint, unmasking a latent ADRB3-S100B program that preserves thermogenesis when lipid fuel is limited. These findings reveal a previously unrecognized role of the SCN in governing thermogenic flexibility and fuel partitioning, and position the ADRB3-S100B axis as a potential target for mitigating circadian misalignment and metabolic disease.

## Introduction

The suprachiasmatic nucleus (SCN), located in the hypothalamus, serves as the master circadian pacemaker, coordinating daily physiological rhythms, including those governing metabolism, by synchronizing peripheral clocks across various metabolic tissues. This temporal coordination is mediated through transcription-translation feedback loops (TTFLs) involving core clock genes such as *Bmal1*, *Per1/2*, and *Cry1/2*, enabling the SCN to integrate environmental cues, primarily light, and drive rhythmic gene expression throughout the body [1–3]. Disruption of SCN function or the TTFLs, whether by genetic alteration or lifestyle factors such as shift work, jet lag, or prolonged light exposure, has been associated with a range of metabolic pathologies, including obesity and diabetes [4–6].

Previous studies have demonstrated that SCN ablation contributes to weight gain and insulin resistance in mice [7] and influences memory processing [8]. Genetic depletion of SCN-regulated clock components, such as *Rev-erbα/β*, significantly impairs metabolic regulation [9,10]. The SCN is also critical for adaptive physiological responses to energy deficits. For example, SCN ablation impairs torpor induction during food deprivation in both hamsters and rats [11,12], indicating its role in energy conservation under metabolic stress. More recently, Tognini and colleagues demonstrated that the SCN exhibits region-specific metabolic plasticity, with exposure to a high-fat diet (HFD) triggering extensive reprogramming of the circadian metabolome, highlighting the SCN's sensitivity to nutritional status [13]. Moreover, our previous work demonstrated that SCN lesioning prevents the hypothermic response typically induced by a 4-hour time-restricted feeding (TRF) window under subthermoneutral environment (STE; 21°C), a physiologically relevant model of combined nutritional and thermal stress [14].

Brown adipose tissue (BAT) is a key thermogenic organ that dissipates energy as heat via uncoupling protein 1 (UCP1), predominantly in response to sympathetic nervous system (SNS) activation. During thermogenesis, BAT relies on fatty acids and glucose as fuel substrates [15–17]. In contrast, white adipose tissue (WAT) primarily functions as a lipid storage depot, mobilizing fatty acids through the activation of hormone-sensitive lipase (HSL) during energy stress. Although both depots are innervated and regulated by the SNS, accumulating evidence suggests that the SCN exerts depot-specific and context-dependent control over their activity, potentially via GABAergic projections that modulate sympathetic tone selectively [18–20]. However, it remains unclear whether and how the SCN regulates substrate preference

SVF, stromal vascular fraction; TG, triglycerides; TH, tyrosine hydroxylase; TRF, time-restricted feeding; TRF-STE, time-restricted feeding in subthermoneutral environments; TTFL, transcription-translation feedback loop; UCP1, uncoupling protein 1; WAT, white adipose tissue; WT, wild-type.

and thermogenic adaptation in peripheral tissues, especially in BAT, under combined nutrient and thermal stress.

One promising candidate in this regulatory axis is S100B, a calcium-binding protein associated with both metabolic and neuronal functions [21]. Elevated circulating levels of S100B have been associated with obesity, insulin resistance, and neurodegeneration in humans [22–24]. S100B has been shown to enhance sympathetic innervation and promote brown adipocyte differentiation, pointing to a potential role in homeostatic thermogenic regulation [25–27]. Beyond its metabolic role, S100B also modulates cell fate by interacting with p53 and inhibiting p21, thereby influencing cellular senescence and proliferation [28,29].

In this study, we investigate the impact of SCN disruption on BAT thermogenesis and metabolic flexibility, employing multiple models of SCN disruption, including surgical lesioning, light-induced circadian arrhythmia, NMDA excitotoxicity, and caspase-3-mediated ablation. We show that SCN disruption impairs lipolysis and adrenergic signaling yet preserves BAT thermogenesis via requiring a compensatory shift from fatty acids to glucose utilization, driven by the ADRB3-S100B axis, which also promotes proliferation and prevents senescence. Interestingly, this adaptation is BAT-specific, as WAT adrenergic responsiveness and lipolysis are diminished. Using the TRF-STE paradigm to simulate nutritional deprivation and thermal stress, we uncover a critical role for the SCN in fuel partitioning and thermogenic adaptation. The identification of the ADRB3-S100B pathway as a glucose-driven thermogenic regulator offers a potential therapeutic strategy for improving metabolic flexibility and overcoming weight-loss resistance.

## Results

### SCN lesioning preserves BAT thermogenesis and disrupts fuel partitioning under TRF-STE

To examine the role of the SCN in thermogenic adaptation under combined nutritional and thermal stress, we implemented a TRF protocol in STE (TRF-STE), a model previously shown to induce hypothermia in wild-type (WT) mice [14]. Mice were restricted to a 4-hour feeding window (ZT16-ZT20) to minimize disruption of SCN-driven circadian rhythmicity [30]. Core body temperature (CBT) was recorded using implanted thermochips, and BAT surface temperature was monitored by infrared thermography. For reference, we reanalyzed previously published CBT data obtained from WT mice housed under conventional housing conditions (CC, 23°C−25°C) [14], where TRF induced only a modest CBT reduction when switched from *ad libitum* to TRF (Fig 1A, dash line). In contrast, under 21°C (STE) conditions, sham-operated mice (sham mice) displayed a much larger drop in CBT and BAT temperature upon food withdrawal at ZT20, consistent with a hypothermic adaptation to fasting (Fig 1A and 1B). By contrast, SCN-lesioned (SCNx) mice (S1A Fig, described in Materials and methods) showed attenuated hypothermia during the second TRF cycle, maintaining significantly sustained CBT and BAT temperatures than sham mice at 21°C or WT under CC (Fig 1A and 1B). To distinguish stress-induced hypothermia from the

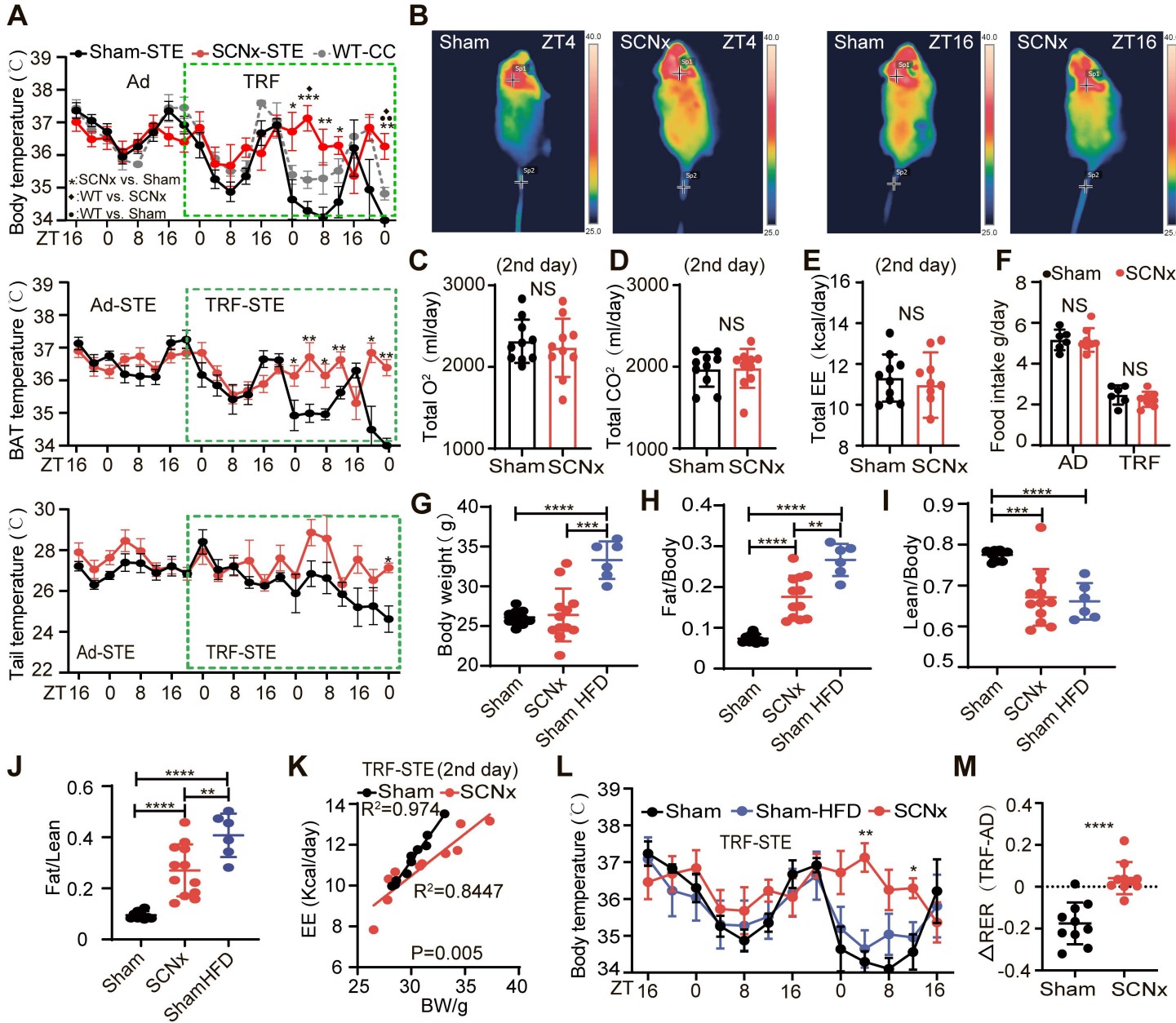

**Fig 1. SCN lesioning preserves BAT thermogenesis and impairs fuel switching under TRF-STE conditions. (A)** Core body temperature (top), including WT mice under conventional conditions CC, 23°C–25°C; dashed line, data reanalyzed from Zhang and colleagues, 2020), interscapular BAT surface temperature (middle), and tail temperature (bottom) in sham or SCNx mice under Ad-STE and TRF-STE. The feeding window (ZT16-ZT20) is indicated by the green shaded area. WT: $n=3$ (reanalyzed from Zhang and colleagues, 2020); sham and SCNx under Ad-STE: $n=10$, sham and SCNx under TRF: $n=5$. All mice were 8 weeks old at the start. After SCN lesioning, mice recovered for 1 week, underwent 1 week of wheel-running screening, and only arrhythmic mice without abrupt body weight gain were selected for 2 weeks of acclimation in 21°C chambers before TRF experiments. Data are shown as mean±SEM. **(B)** Representative infrared thermographic images showing surface body temperature at ZT4 (rest phase) and ZT16 (active phase) under TRF-STE in sham and SCNx mice. **(C–E)** Measurements of total $VO_2$ **(C)**, total $VCO_2$ **(D)**, and total EE **(E)** on day 2 of TRF-STE ($n=10$ per group). **(F)** Food intake analysis under Ad-STE and TRF-STE in sham ($n=6$) and SCNx mice ($n=8$). **(G–J)** Body composition analysis of fat mass and lean mass in sham ($n=10$), SCNx ($n=12$), and HFD-fed mice ($n=6$). **(K)** GLM analysis of EE with group as the independent variable and body weight as the covariate under TRF-STE conditions ($n=10$ per group). **(L)** CBT in sham ($n=5$), HFD-fed ($n=4$), and SCNx ($n=5$) mice during TRF-STE. Data are mean±SEM. **(M)** ΔRER at ZT16 following the transition from Ad-STE to TRF-STE in sham and SCNx mice ($n=10$ per group). Unless otherwise indicated, data are presented as mean±SD. NS: not significant, $*p<0.05$, $**p<0.01$, $***p<0.001$, and $****p<0.0001$. Tests used: unpaired two-tailed Student $t$ test **(C–J**, and **M)**, and two-way ANOVA with Sidak'S multiple comparisons test **(A, L)**. The data underlying the graphs shown in the figure can be found in S1 Source Data.

normal sleep-associated temperature dip, we analyzed the TRF at different circadian phases (ZT4-8, ZT10-14, and ZT22-2). Hypothermia coincided consistently with food deprivation across phases, independently of sleep onset (S1B Fig), indicating that SCN lesioning prevents adaptive hypothermia rather than abolishing the physiological circadian trough. Tail temperature, a proxy for peripheral vasodilation, remained unchanged between groups (Fig 1A).

To further examine whole-body energy metabolism, we assessed oxygen consumption ($VO_2$) and energy expenditure (EE) via indirect calorimetry. Sham and SCN-lesioned mice maintained comparable $VO_2$ and EE under both *ad libitum* (Ad-STE) (S1C–S1F Fig) and TRF-STE conditions (Figs 1C–1E and S1G), with no significant differences in food intake (Fig 1F). However, body composition analysis revealed an increase in fat mass and a concomitant reduction in lean mass in SCN-lesioned mice despite similar body weight (Fig 1G–1J). To determine whether SCN lesioning alters EE independently of body mass, we applied a General Linear Model (GLM) with body weight as a covariate. A significant group × mass interaction violated the assumptions required for analysis of covariance (ANCOVA), but GLM still identified a significant main effect of group on EE (Figs 1K and S1H). These results indicate that SCN lesioning reduces EE.

To test whether increased fat deposition could offset the elevated thermogenic demand imposed by TRF-STE, sham mice were fed a HFD prior to exposure. Despite marked increases in body weight and adiposity (Fig 1G–1J), HFD-fed mice still exhibited a significant drop in CBT during TRF-STE (Fig 1L). These results indicate that increased fat mass alone is insufficient to counteract hypothermia under combined nutritional and thermal stress, supporting a key role in SCN-mediated control of thermogenic adaptation.

SCN-lesioned mice displayed significantly lower locomotor activity under Ad-STE and TRF-STE (S1I and S1J Fig), which may partially account for their decreased EE (Figs 1K and S1H). However, their ability to maintain body temperatures despite reduced EE suggests the presence of a compensatory mechanism in energy metabolism. To investigate whether this compensation involved altered substrate utilization, we examined changes in respiratory exchange ratio (ΔRER) from Ad-STE to TRF-STE transition as an indicator of fuel preference. Under TRF-STE, sham mice displayed the expected shift from carbohydrate to lipid oxidation (Fig 1M), reflecting adaptive metabolic flexibility. In contrast, SCN-lesioned mice failed to make this transition, maintaining a glucose-dependent profile (Fig 1M). This impaired shift in fuel utilization indicates a loss of metabolic flexibility, likely attributable to impaired SCN function and decreased lipid mobilization.

## SCN lesioning impairs lipid mobilization but enhances glucose-driven thermogenesis in BAT under TRF-STE

To investigate the mechanisms underlying preserving thermogenesis in SCN-lesioned mice under combined nutrient and thermal stress, we assessed tissue-specific glucose uptake and metabolic flexibility. Utilizing positron emission tomography/computed tomography (PET/CT) with $^{18}$F-fluorodeoxyglucose ($^{18}$F-FDG) at ZT6 after a 10-hour fast, we observed a significant increase in glucose uptake in SCN-lesioned BAT compared to sham controls, a finding corroborated by gamma counting (Fig 2A–2C). In contrast, glucose uptake in WAT was significantly reduced, while hepatic uptake remained unchanged (Fig 2C), suggesting a selective redistribution of glucose toward BAT under SCN disruption and a metabolic shift favoring glucose as a thermogenic fuel.

To functionally validate this substrate switch, we performed extracellular flux analysis on ex vivo BAT explants. Upon glucose stimulation (10 mM), BAT from SCN-lesioned mice exhibited a significantly increased extracellular acidification rate (ECAR), indicating enhanced glycolytic flux (Fig 2D and 2E). These results are consistent with the elevated RER observed during TRF-STE (Fig 1M), further supporting a glucose-driven thermogenic program in the absence of SCN regulation.

Consistent with these functional changes, SCN-lesioned BAT showed upregulated expression of glycolytic enzymes such as *Pfkl* and *Pkm*, indicating enhanced glucose catabolism (Fig 2F). Additionally, *Lctl*, a gene involved in alternative carbohydrate metabolism (e.g., lactose and galactose metabolism) [31], was also upregulated (Fig 2F), suggesting broader transcriptional reprogramming of carbohydrate metabolism in response to SCN disruption.

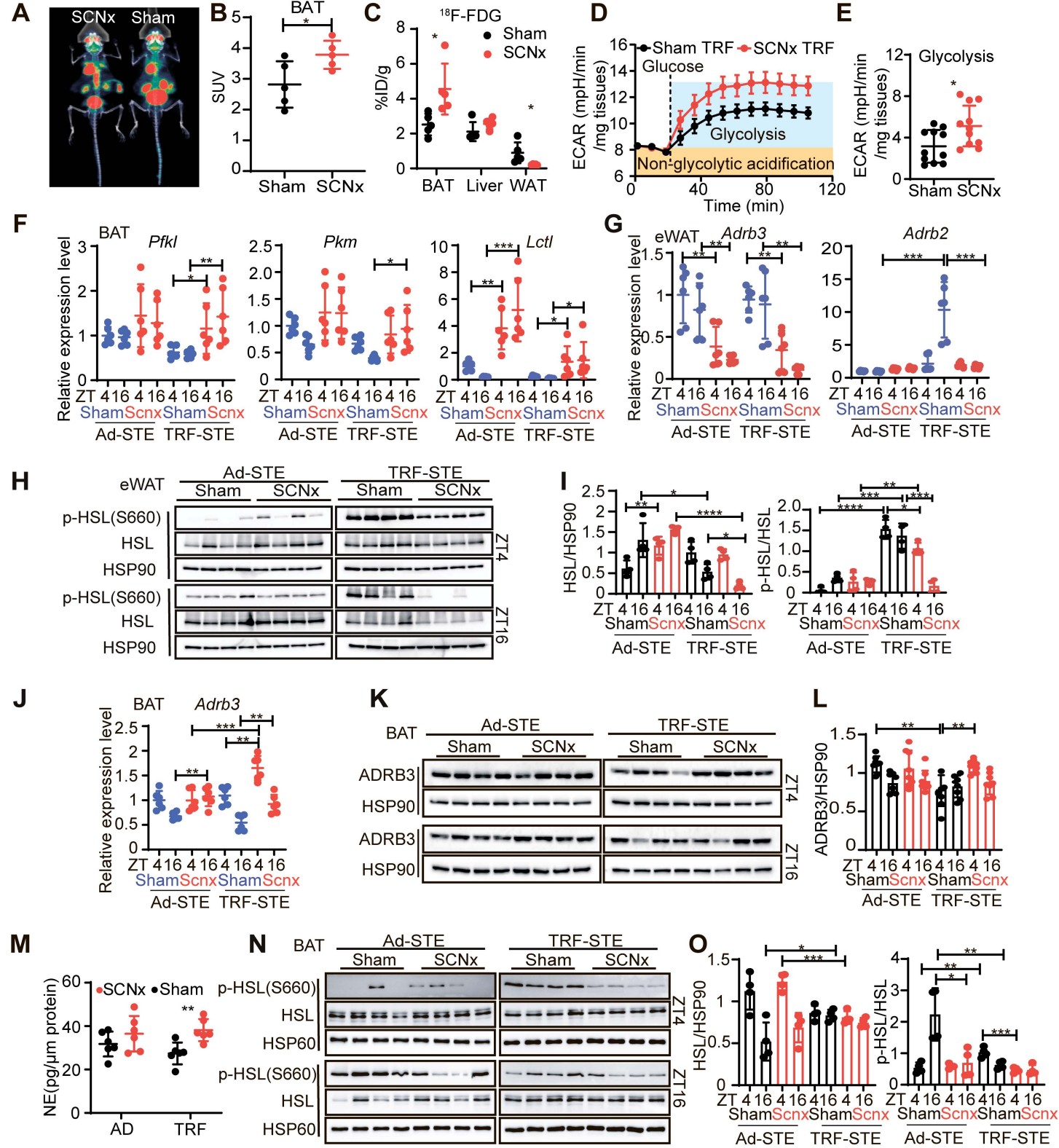

**Fig 2. SCN lesioning impairs lipid mobilization and metabolic flexibility under TRE-STE. (A–C)** [18]F-FDG PET-CT imaging and glucose uptake quantification. Representative PET-CT images 60 min after intravenous injection of [18]F-FDG under TRF-STE **(A)**. Quantification of [18]F-FDG uptake

across tissues by PET-CT **(B)**. Radioactivity measurements in interscapular BAT, liver, and eWAT using gamma counting **(C)**. $n = 5$ per group. **(D and E)** Glycolytic flux in interscapular BAT explants. Seahorse extracellular flux of ECAR in interscapular BAT explants from sham and SCNx mice under TRF-STE at ZT16 **(D)**. Quantification of basal glycolysis, normalized to tissue weight **(E)**. $n = 11$ per group. **(F)** QPCR analysis of glycolytic and carbohydrate metabolism genes (*Pfkl*, *Pkm*, and *Lctl*) in interscapular BAT from sham and SCNx mice under Ad-STE and TRF-STE ($n = 6$ per group). **(G)** Expression of adrenergic receptors (*Adrb3*, *Adrb2*) in eWAT from sham and SCNx mice under Ad-STE and TRF-STE ($n = 6$ per group). **(H and I)** Western blot analysis of HSL and phosphor-HSL (HSL Ser660) in eWAT from sham and SCNx mice under Ad-STE and TRF-STE **(H)**, with densitometry analysis **(I)**. $n = 4$ per group. **(J)** Expression of *Adrb3* in interscapular BAT from sham and SCNx mice under Ad-STE and TRF-STE ($n = 6$ per group). **(K and L)** Western blot **(K)** and densitometry analysis **(L)** of ADRB3 levels in interscapular BAT from sham and SCNx mice under Ad-STE and TRF-STE ($n = 8$ per group). **(M)** NE levels in interscapular BAT measured by ELISA from sham and SCNx mice at ZT16 under Ad-STE and TRF-STE ($n = 6$ per group), normalized to total protein. **(N and O)** Western blot analysis of HSL and phosphor-HSL (HSL Ser660) in interscapular BAT from sham and SCNx mice under Ad-STE and TRF-STE **(N)**, with densitometry analysis **(O)**. $n = 4$ per group. All data are presented as mean ± SD. $*p < 0.05$, $**p < 0.01$, $***p < 0.001$, and $****p < 0.0001$ as determined by unpaired two-tailed Student $t$ test (**B, C, E-G, I, J, L, M,** and **O**). The data underlying the graphs shown in the figure can be found in S1 Source Data. Raw blot images can be found in S1 Raw Images.

Given the crucial role of β3-adrenergic signaling in the regulation of both thermogenesis and lipolysis [15,32,33], we assessed sympathetic signaling in WAT. In epididymal WAT (eWAT), expressions of *Adrb2* and *Adrb3* were significantly reduced in SCN-lesioned mice under TRF-STE (Fig 2G), consistent with indicative of suppressed sympathetic tone. Functionally, this was reflected by a failure to induce phosphorylation of HSL at Ser660, a key activation site downstream of extracellular signal-regulated kinase (ERK), suggesting impaired lipolysis in SCN-lesioned eWAT in response to TRF-STE (Figs 2H, 2I, and S2A).

Conversely, in BAT, both *Adrb3* mRNA and ADRB3 protein levels were elevated at ZT4 and remained sustained at ZT16 under TRF-STE in SCN-lesioned mice, in contrast to their downregulation in sham controls (Fig 2J–2L). Norepinephrine (NE) levels in BAT were also significantly higher in SCN-lesioned mice (Fig 2M), suggesting preserved or even enhanced β3-adrenergic signaling in BAT under TRF-STE. However, despite increased *Adrb3* expression, HSL phosphorylation remained low in SCN-lesioned BAT under TRE-STE (Fig 2N and 2O), indicating a disconnect between receptor abundance and downstream lipolytic activation and accessibility of fatty acids to mitochondrial oxidation. Consistent with this, genes involved in fatty acid mobilization and oxidation, such as *Cyp2e1*, *Irf4*, and *Cyp4a10*, were significantly downregulated in SCN-lesioned BAT (S2B Fig). H&E staining and Oil Red O staining further revealed the presence of larger lipid droplets in SCN-lesioned BAT under both Ad-STE and TRF-STE, consistent with impaired fatty acid utilization (S2C and S2D Fig).

Together, these findings demonstrate that SCN lesioning drives glucose-driven thermogenesis in BAT, while simultaneously impairing lipid mobilization in eWAT and BAT, likely due to disrupted sympathetic regulation. The apparent dissociation between β3-adrenergic receptor expression and lipolytic activation highlights compensatory metabolic reprogramming in BAT under SCN disruption, whereby it switches from using lipid to carbohydrate substrates to sustain thermogenesis.

## SCN modulates thermogenic capacity via sympathetic signaling

To elucidate the anatomical and functional connection between the SCN and BAT, we performed anterograde tracing using HSV-eGFP injected into the SCN, which identified GFP-positive axonal terminals in interscapular BAT. Complementary retrograde tracing with PRV-CAG-GFP revealed SCN-positive neurons projecting to interscapular BAT, establishing an anatomical link between the SCN and BAT (Figs 3A, 3B, and S3A) [34].

To investigate whether the SCN-mediated regulation of BAT thermogenesis operates via sympathetic innervation, we first examined tyrosine hydroxylase (TH), a marker of catecholaminergic sympathetic activity. Although total TH protein levels in BAT were unchanged between groups (S3B and S3C Fig), Adipo-Clear 3D imaging revealed a marked increase in sympathetic fiber density in SCN-lesioned BAT compared to sham controls (Fig 3C and 3D). To functionally validate the necessity of local sympathetic input, we selectively ablated catecholaminergic fibers in BAT using 6-hydroxydopamine (6-OHDA). Effective denervation was confirmed by the significant reduction in TH signal (S3D and S3E Fig). Under

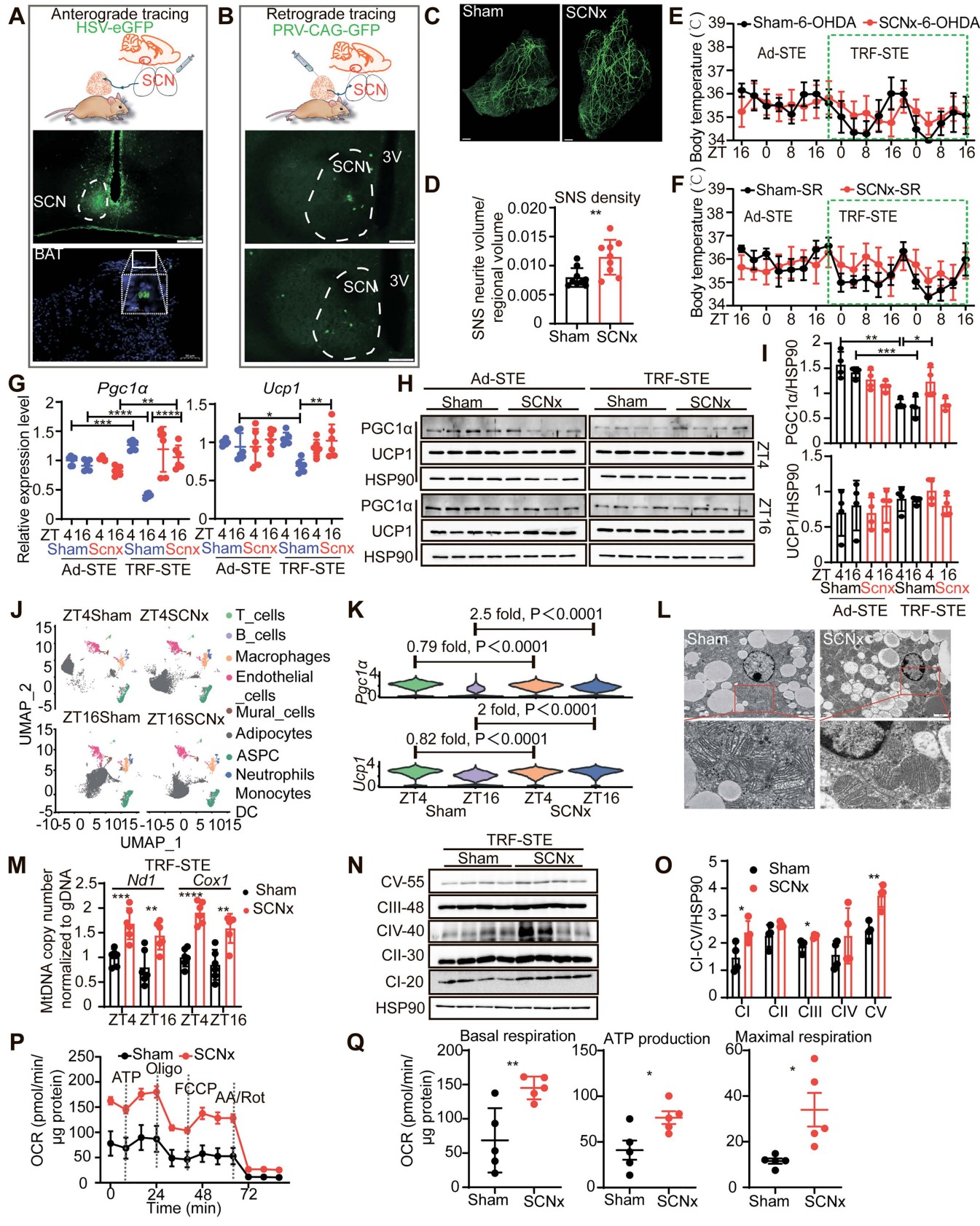

**Fig 3. SCN regulates thermogenic capacity via sympathetic signaling. (A)** Schematic representation of anterograde tracing and an image showing HSV-eGFP-labeled axonal projections from the SCN terminating in the interscapular BAT region. **(B)** Schematic and image showing the retrograde tracing from interscapular BAT to SCN, visualizing tracing signals in the SCN. **(C and D)** 3D whole-mount TH immunostaining (**C**) and quantification of sympathetic nerve fiber density (**D**) in interscapular BAT from sham and SCNx mice under TRF-STE from 10 regions ($n = 1$ per group). **(E)** Body temperature response to 6-OHDA treatment in sham and SCNx mice from Ad-STE to TRF-STE. Data are mean ± SEM, $n = 6$ per group, from the same experiments as interscapular BAT and tail temperature as S3F and S3G Fig. **(F)** Thermal response to SR59230A treatment in sham ($n = 6$) and SCNx mice ($n = 5$) from Ad-STE to TRF-STE. Data are mean ± SEM. From the same experiments as interscapular BAT and tail temperature as S3H and S3I Fig. SR: SR59230A. **(G)** Relative mRNA expression of *Pgc1α* and *Ucp1* at ZT4 and ZT16 under Ad-STE and TRF-STE ($n = 6$ per group). **(H and I)** Western blot analysis of PGC1α and UCP1 levels in interscapular BAT from sham and SCNx mice **(H)**, with densitometry analysis **(I)**. $n = 4$ per group. **(J)** UMAP plot showing snRNA-seq data and clustering into 8 major cell types. **(K)** Expression of *Pgc1α* and *Ucp1* within adipocyte clusters at ZT4 and ZT16 in sham vs. SCNx mice; exact $p$-values reported. **(L)** Electron microscopy images showing mitochondria in interscapular BAT from sham and SCNx mice under TRF-STE conditions. The red square highlights a magnified view of the mitochondrial structure. **(M)** Quantification of mitochondrial DNA content in interscapular BAT from sham and SCNx mice under TRF-STE ($n = 6$ per group). **(N and O)** Western blot analysis of the OXPHOS in interscapular BAT from sham and SCNx mice under TRF-STE conditions **(N)**, with densitometry analysis **(O)**. $n = 4$ per group. **(P)** OCR was measured in mitochondria isolated from interscapular BAT of sham and SCNx mice using Seahorse XF Analyzer. Measurements were taken under basal conditions and after sequential injection of oligomycin (ATP synthase inhibitor), FCCP (mitochondrial uncoupler), and a combination of rotenone/antimycin A (complex I/III inhibitors of the electron transport chain). Data are mean ± SEM, $n = 5$ per group. **(Q)** Quantification of basal respiration, ATP-linked respiration, and maximal respiration. $n = 5$ per group. Unless otherwise indicated, data are presented as mean ± SD. *$p < 0.05$, **$p < 0.01$, ***$p < 0.001$, and ****$p < 0.0001$ as determined by unpaired two-tailed Student $t$ test (**D, G, I, M, O**, and **Q**), two-way ANOVA with Sidak'S multiple comparisons test **(E and F)**. Scale bars, 400 μm **(C)**, 200 μm **(A, middle)**, 100 μm **(B)**, 50 μm (bottom of **A**), 10 μm (magnified view of **A**), 2 μm (**L**, upper), 500 nm (**L**, bottom). Schematic (**A, B**) created in BioRender.com. The data underlying the graphs shown in the figure can be found in S1 Source Data. Raw blot images can be found in S1 Raw Images.

TRF-STE conditions, 6-OHDA treatment resulted in thermogenic responses that were indistinguishable between sham and SCN-lesioned mice, indicating that intact sympathetic innervation is required for SCN-dependent BAT activation (Figs 3E, S3F, and S3G). To further assess the downstream effector mechanisms, we pharmacologically blocked β3-adrenergic signaling using SR59230A, a selective ADRB3 antagonist. Following SR59230A administration, sham and SCN-lesioned mice exhibited comparable core and BAT temperatures under TRF-STE (Figs 3F, S3H, and S3I), supporting the conclusion that β3-adrenergic signaling is a critical mediator of SCN-dependent thermogenic regulation.

We next examined the expression of canonical thermogenic markers, *Ppargc1a* (*Pgc1α*) and *Ucp1* in BAT. In sham mice, TRF-STE intensified the diurnal variation in these markers, with notably reduced expression at ZT16. In contrast, SCN-lesioned mice maintained expression of both markers at ZT16, with PGC1α showing a more robust and sustained response than UCP1 (Fig 3G–3I), suggesting a failure to suppress thermogenesis in the absence of SCN regulation.

In order to analyze the cellular heterogeneity and the dynamic expression characteristics of thermogenic genes (*Pgc1α*, *Ucp1*) in specific cell populations in BAT, we performed single-cell nuclear RNA sequencing (snRNA-seq) of BAT from sham and SCN-lesioned mice at ZT4 and ZT16 under TRF-STE conditions. Clustering analysis identified eight major clusters, ranging from 100 to 9,500 cells per cluster, with each expressing *Pgc1α* and *Ucp1* (Figs 3J and S3J–S3L, S1, and S2 Tables). In brown adipocytes of SCN-lesioned mice, expression of both genes was slightly lower at ZT4 but significantly elevated at ZT16 compared to sham controls (Fig 3K). These findings confirm that SCN lesioning prevents the physiological TRF-STE-mediated suppression of *Pgc1α* and *Ucp1* expression, so that defects in SCN results in sustained thermogenic activation in brown adipocytes.

SCN-lesioned BAT displayed increased mitochondrial density, enhanced lamellar cristae structure (Fig 3L), and a 1.5-fold increase in mitochondrial DNA (mtDNA) content (Fig 3M), along with upregulation of oxidative phosphorylation complex proteins (OXPHOS) (Fig 3N and 3O). To determine functional mitochondrial activity, we measured oxygen consumption rate (OCR) using Seahorse metabolic flux analysis. BAT mitochondria from SCN-lesioned mice showed elevated basal respiration, increased OCR in response to oligomycin and FCCP (Fig 3P and 3Q), indicating enhanced mitochondrial capacity and uncoupling potential under SCN disruption. To test whether this effect depended on sympathetic input, BAT was treated with 6-OHDA prior to respiration assays. Following treatment with 6-OHDA, OCR remained at comparable levels in both sham and SCN-lesioned mice (S3M and S3N Fig).

PLOS Biology

Collectively, these findings demonstrate that the SCN exerts critical control over BAT thermogenic capacity by modulating sympathetic outflow. Disrupted SCN results in activation β3-adrenergic signaling, and promotes mitochondrial biogenesis and function to prevent the normal adaptive decreased thermogenic response under TRF-STE conditions.

**SCN lesioning reverses the TRF-STE induced BAT senescence and promotes cell proliferation via S100B**

To investigate the molecular mechanism underlying SCN-regulated BAT thermogenesis, we performed RNA sequencing (RNA-seq) on interscapular BAT from sham and SCN-lesioned mice at ZT4 and ZT16 under Ad-STE and TRF-STE. Among the 25,240 genes analyzed, SCN-lesioned mice showed 77 upregulated and 24 downregulated genes at ZT4 and 316 upregulated and 274 downregulated genes at ZT16 compared to sham controls under TRF-STE (fold change > 2, $p < 0.01$) (Fig 4A). SCN lesioning significantly downregulated genes associated with fatty acid oxidation (*Irf4*, *Cyp2e1*, *Cyp4a10*, *Cyp2b10*, and *Cyp2f2*) and lipid transport (*Mfsd2a*, *Slc7a8*, and *Slc43a1*), while upregulating genes involved in lipogenesis and lipid storage (*Amd1*, *Mest*, *Gdf15*, *Fgf18*, *Ptgds*, and *Hpgds*) as well as carbohydrate metabolism (*Pfkl*, *Pkm*, and *Lctl*) (S4A Fig). These transcriptomic changes suggest impaired lipid mobilization alongside enhanced lipid accumulation on one side and increased glycolytic activity in BAT from SCN-lesioned mice, consistent with our earlier observations of decreased lipolysis and increased glucose utilization (Fig 2).

Notably, SCN lesioning reversed the TRF-STE-mediated suppression of proliferation-related genes such as *S100b*, *Ccnd1*, *Prelp*, and *Top2a*, which were physiologically downregulated in sham mice at both time points (ZT4 and ZT16) (Fig 4A). Moreover, genes involved in methylation and detoxification (*Ahcy*, *Gnmt*, *Pxmp2*, and *Cyp2e1*) were consistently downregulated in SCN-lesioned mice (Figs 4A and S4B), suggesting a possible metabolic reallocation away from auxiliary pathways toward inappropriately activated thermogenic and proliferative programs.

Ingenuity Pathway Analysis (IPA) revealed significant enrichment of circadian rhythm, erythropoietin signaling, iron homeostasis, and S100 family signaling pathways, the latter of which showed significant activation (z-scores: −2.111 at ZT4, −2.143 at ZT16) in SCN-lesioned BAT (Fig 4B, S3 Table; IPA: $p < 0.01$). Complementary KEGG and Gene Ontology analyses showed that the S100 signaling intersects with GPCR signaling, angiogenesis, and cAMP/adrenergic pathways (Fig 4C), indicating a contributing role in adaptive thermogenic responses.

Under TRF-STE, sham mice displayed upregulation of senescence-associated genes such as *Cdkn1a* (*p21*) and *Gadd45γ*, especially at ZT16 following prolonged fasting (S4C Fig), suggesting an energy-conserving senescence response. In contrast, SCN lesioning suppressed this physiological senescence response and maintained elevated expression of *S100b* and *Ccnd1*, which were repressed in sham controls under TRF-STE (S4D Fig).

S100B, a calcium-binding protein with known roles in cell signaling, proliferation, and stress adaptation [28], was positively correlated with proliferation markers (*Trp53*, *Mki67*, *Ccnd1*, *Top2a*, *Cd34*, and *Prelp*) and inversely correlated with senescence markers (*Cdkn1a* and *Gadd45γ*) based on RNA-seq data (Fig 4D and 4E). In SCN-lesioned mice under TRF-STE, the increase in S100B was associated with enhanced expression of proliferation markers and suppression of senescence-associated genes (S4E and S4F Fig), suggesting that SCN activity modulates BAT cellular plasticity via S100B-dependent mechanisms.

To directly assess proliferation, 5-ethynyl-2′-deoxyuridine (EdU) labeling revealed significantly increased EdU⁺ nuclei in BAT from SCN-lesioned mice under both Ad-STE and TRF-STE conditions compared to sham controls (Fig 4F and 4G). Consistently, anti-PDGFRα immunostaining of BAT sections showed an increased number of PDGFRα⁺ preadipocytes in SCN-lesioned mice relative to sham mice under both Ad-STE and TRF-STE conditions (S4G and S4H Fig). In vitro, recombinant S100B promoted proliferation of PDGFRα⁺ preadipocytes (Fig 4H–4K) and accelerated early expression of brown adipocyte differentiation markers (Fig 4L), indicating dual roles in precursor expansion and thermogenic commitment. In parallel, senescence-associated β-galactosidase (SA-β-gal) staining showed a marked increase in senescent cells in sham BAT under TRF-STE, which was significantly reduced in SCN-lesioned BAT (Fig 4M). Together, these findings support a model in which SCN-regulated S100B expression protects BAT from stress-induced senescence and enhances its regenerative and thermogenic capacity under nutrient and thermal stress.

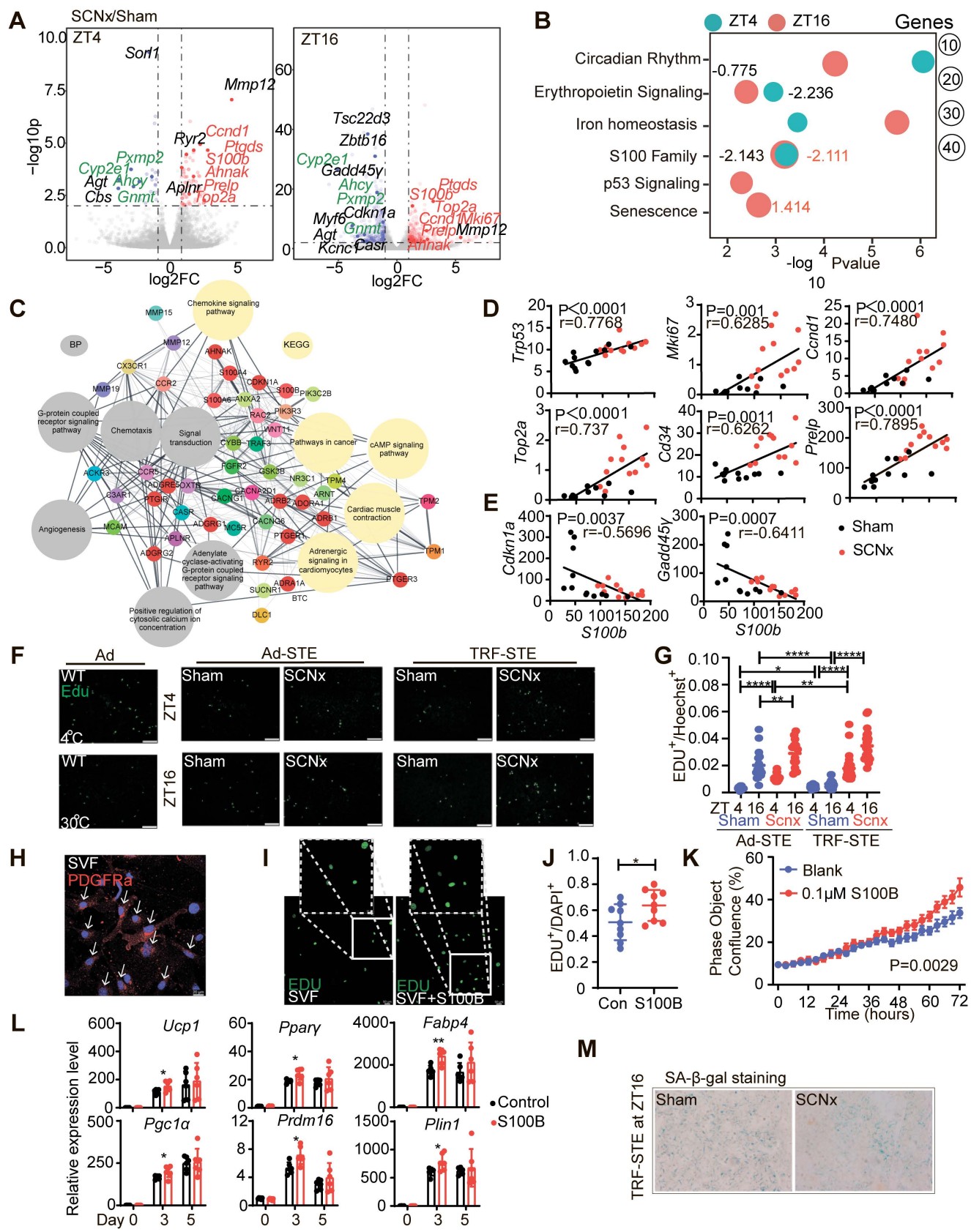

**Fig 4. SCN lesioning counteracts TRF-STE-induced BAT senescence via S100B. (A)** Volcano plot showing differentially expressed genes (DEGs) in interscapular BAT from sham and SCNx mice under TRF-STE. DEGs were defined by fold change > 2, $p < 0.01$. Commonly upregulated genes are highlighted in red and downregulated genes in green at both ZT4 and ZT16. $n = 3$ per group. **(B)** IPA of DEGs at ZT4 and ZT16. Pathways with significant enrichment ($p < 0.01$) are shown, with activation z-scores indicated for each pathway. **(C)** DAVID pathway analysis of S100 family-associated DEGs. Enriched KEGG pathways are shown in yellow, and Gene Ontology biological processes (BP) are shown in gray. **(D and E)** Correlation analysis of RNA-seq-derived *S100b* expression with cell proliferation markers **(D)** and senescence-related genes **(E)** in sham and SCNx mice. Pearson correlation coefficients and *p*-values are shown. **(F and G)** Cell proliferation analysis via EdU incorporation. Representative images showing EdU incorporation in interscapular BAT from sham and SCNx mice at ZT4 and ZT16 under Ad-STE and TRE-STE with 4°C and 30°C as control **(F)**. Quantification of EdU+ nuclei as a percentage of total (Hoechst-stained) nuclei **(G)**. $n = 5$ per group. **(H–K)** Assessment of S100B-induced proliferation in PDGFRα+ preadipocytes. PDGFRα-positive stromal vascular fraction (SVF) cells **(H)**, representative images of EdU-positive cells treated with recombinant S100B **(I)**, statistical analysis of EdU+ nuclei **(J)**, and cell growth curves of control vs. S100B-treated preadipocytes **(K)**. **(L)** Expression of differentiation-related genes during SVF from interscapular BAT induced to differentiate into mature adipocytes. $n = 6$ per group. **(M)** Representative β-galactosidase staining images showing senescent cell abundance in interscapular BAT from sham and SCNx mice under TRF-STE. Data are presented as mean ± SD. Statistical significance was determined using unpaired two-tailed Student *t* test **(G, J,** and **L)**, Pearson correlation analysis **(D and E)**, and two-way ANOVA with Sidak'S multiple comparisons test **(K)**. *$p < 0.05$, **$p < 0.01$ and ****$p < 0.0001$. Scale bars, 100 μm **(F and M)**, 50 μm **(I)**, 20 μm **(H** and magnified view of **I)**. The data underlying the graphs shown in the figure can be found in S1 Source Data.

## S100B functions as a nutrient-sensitive effector that amplifies β3-adrenergic signaling and thermogenic capacity in BAT

S100B emerged from our transcriptomic analyses as a potential effector of thermogenic plasticity under conditions of SCN inactivation and metabolic stress. To assess its responsiveness to energetic states, we measured *S100b* expression in interscapular BAT under diverse nutritional and thermal conditions. A 48-hour fast completely suppressed *S100b*, whereas HFD feeding significantly increased its expression (Fig 5A). Thermoneutral acclimation (30°C) markedly suppressed *S100b*, while cold exposure (4°C) failed to induce *S100b*, despite robust *Ucp1* upregulation (Fig 5B). These findings demonstrate that *S100b* is regulated by nutrient availability rather than cold stress. Furthermore, serum concentrations of S100B, triglycerides (TG), and non-esterified fatty acids (NEFA) were unchanged between SCN-lesioned and sham mice (Fig 5C–5E), indicating that *S100b* regulation occurs in a tissue-autonomous manner within BAT, independent of systemic metabolic cues. This finding contrasts with prior reports associating circulating S100B with obesity and neuroinflammation [22] and underscores the localized nature of SCN-mediated control of S100B in BAT.

Histological analysis confirmed reduced S100B expression in BAT of sham mice under TRF-STE, whereas SCN-lesioned mice maintained high levels (Fig 5F). snRNA-seq confirmed this result, revealing elevated *S100b* expression specifically in brown adipocytes of SCN-lesioned mice at both ZT4 and ZT16, with no significant changes detected in other cell types (S5A Fig and S5B, S4 Table). Western blot analyses confirmed corresponding increases in S100B and CCND1 proteins, alongside decreased levels of the senescence marker p21 in SCN-lesioned BAT (Fig 5G and 5H).

To determine whether β3-adrenergic signaling regulates S100B expression and function, we treated mice with SR59230A. This intervention eliminated the expression differences in *S100b*, *Ccnd1*, and *Cdkn1a* between SCN-lesioned and sham mice (Fig 5I), indicating that ADRB3 signaling modulates the S100B-mediated balance between proliferation and senescence. These results are consistent with our earlier observations showing that SR59230A inhibits SCN-lesion-induced BAT thermogenesis (Fig 3F).

To evaluate the possibility of a direct molecular interaction between S100B and ADRB3, we employed ColabFold protein modeling, which predicted a high-confidence interaction localized to the intracellular domain of ADRB3 (Fig 5J and 5K). This interaction was experimentally validated by co-immunoprecipitation (coIP), confirming physical association between the two proteins (Fig 5L). Modulation of *S100b* expression in BAT altered ADRB3 protein abundance and electrophoretic mobility (Fig 5M and 5N), though this was not attributable to differences in protein degradation (S5C Fig), suggesting that S100B enhances receptor sensitivity rather than stability, as previously proposed [35,36].

To assess the functional consequences of this interaction, we treated primary brown adipocytes (PPDIVs) with recombinant S100B. This significantly increased ECAR, indicative of enhanced glycolytic metabolism (Fig 5O and 5P).

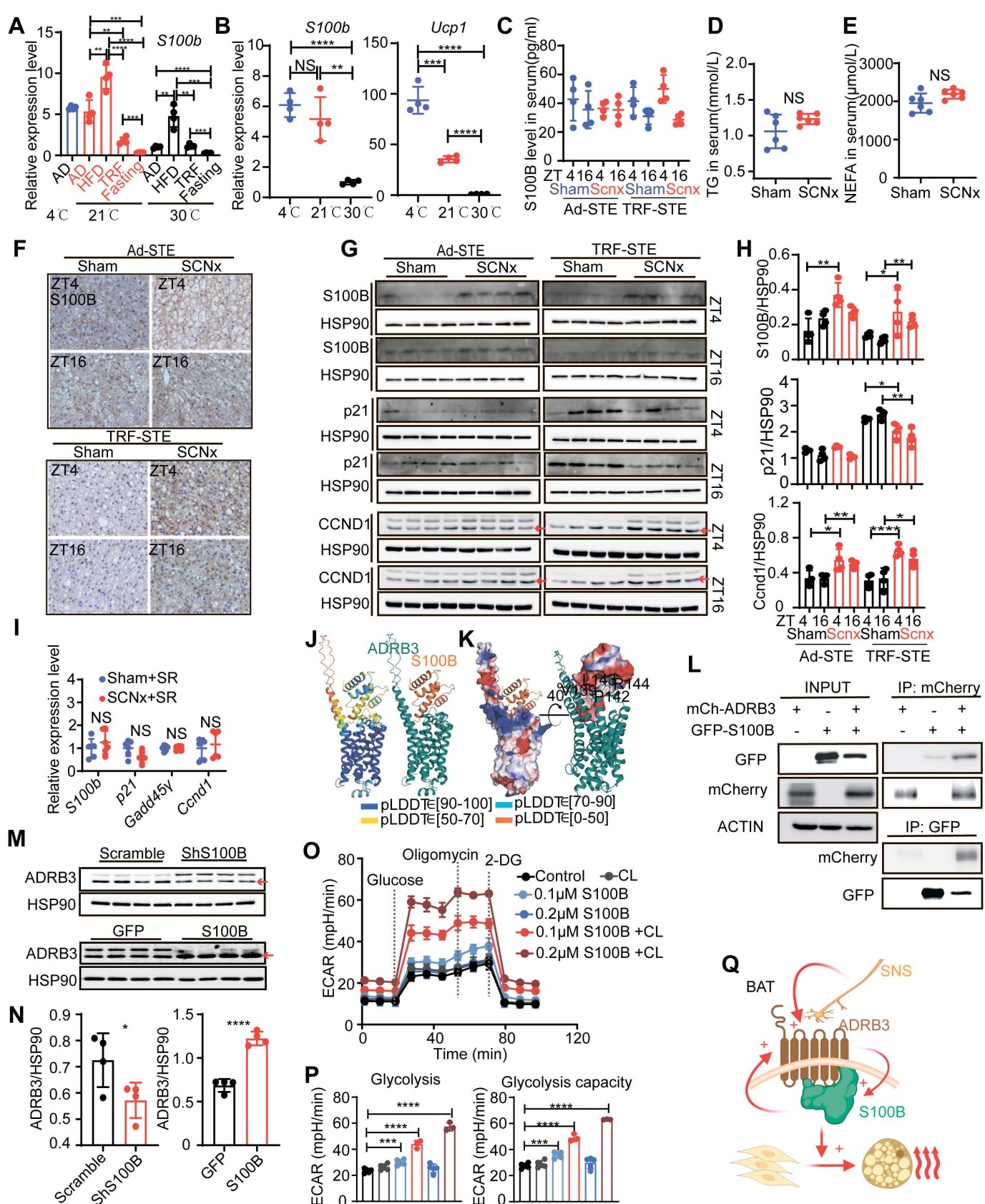

**Fig 5. S100B senses nutrients and amplifies β3-adrenergic signaling. (A)** *S100b* expression levels under various nutritional states (fasting, HFD, and TRF) and environmental temperatures (4°C, 21°C, and 30°C). *n* = 4 per group. **(B)** *S100b* and *Ucp1* expression levels in indicated temperatures. *n* = 4 per group. **(C)** Serum S100B levels were measured by ELISA in sham and SCNx mice under Ad-STE and TRF-STE at ZT4 and ZT16. *n* = 4 per group. **(D and E)** Serum biochemical assays showing levels of TG **(D)** and NEFA **(E)** in sham and SCNx mice under Ad-STE (*n* = 6 per group). **(F)** Representative immunohistochemistry images of S100B protein in interscapular BAT from sham and SCNx mice under Ad-STE and TRF-STE conditions. Scale bars, 50μm. **(G and H)** Western blot analysis **(G)** and densitometric quantification **(H)** of S100B, p21, and CCND1 protein levels in interscapular BAT from sham and SCNx mice at ZT4 and ZT16. *n* = 4 per group. Red arrows indicate quantified bands. **(I)** Relative mRNA levels of indicated genes in interscapular BAT from sham and SCNx BAT following ADRB3 antagonist with SR59230A. *n* = 6 per group. SR: SR59230A. **(J and K)** Structural prediction of the human ADRB3-S100B complex using AlphaFold2/ColabFold. The human ADRB3/S100B complex model **(J)** was predicted by ColabFold and colored in blue, cyan, yellow, and orange according to different prediction confidence (pLDDT, predicted local distance difference test). Surface electrostatic representation showing ADRB3 (deep teal) and S100B (orange) with positive and negative charges indicated in blue and red, respectively **(K)**. **(L)** coIP of ADRB3 and S100B in HEK293T cells. **(M and N)** Western blot **(M)** and densitometry analysis **(N)** of ADRB3 protein levels in interscapular BAT following *S100b* knockdown or overexpression in vivo. *n* = 4 per group. Red arrows indicate quantified bands. **(O and P)** ECAR analysis **(O)** and quantification of basal glycolysis and glycolytic capacity **(P)** in primary preadipocytes treated with S100B and/or β3-agonist CL-316243. Glucose, oligomycin, and 2-DG were sequentially injected. Data are presented as mean ± SEM. Control, CL, 0.1 μM S100B and 0.2 μM S100B: *n* = 6; 0.1 μM S100B + CL and 0.2 μM S100B+CL: *n* = 3. **(Q)** Schematic model illustrating the proposed ADRB3-S100B signaling axis. SCN lesioning enhances SNS activity, thereby promoting ADRB3 signaling and upregulating S100B expression. S100B, in turn, increases ADRB3 sensitivity, establishing a positive feedback loop that sustains thermogenesis and stimulates preadipocyte proliferation. Created in BioRender.com. Unless otherwise indicated, data are presented as mean ± SD. NS: not significant, *p < 0.05, **p < 0.01, ***p < 0.001, and ****p < 0.0001 by unpaired two-tailed Student *t* test (**A**–**E, H, I, N,** and **P**). The data underlying the graphs shown in the figure can be found in S1 Source Data. Raw blot images can be found in S1 Raw Images.

Co-treatment with the β3-agonist CL-316243 synergistically amplified ECAR, indicating that S100B potentiates β3-adrenergic signaling to promote glucose-dependent thermogenesis (Fig 5O and 5P).

Together, these results position S100B as a nutrient-sensitive, SCN-regulated effector of BAT thermogenic plasticity. Acting through the ADRB3-S100B axis, SCN signaling integrates sympathetic tone with transcriptional and metabolic programs that enhance glycolysis, stimulate preadipocyte proliferation, and repress senescence. This positive feedback loop supports sustained BAT thermogenesis under combined nutritional and thermal stress (TRF-STE), and highlights S100B as a critical molecular node in SCN-mediated metabolic adaptation (Fig 5Q).

## S100B is both necessary and sufficient to mediate SCN-dependent thermogenic resilience in BAT

To determine whether S100B is functionally required for SCN-mediated thermogenesis, we selectively knocked down *S100b* in the BAT of SCN-lesioned mice using AAV-U6-shRNA(*S100b*)-EGFP-P2A, with a scrambled shRNA vector as control (Fig 6A). Knockdown of *S100b* significantly increased expression of the senescence marker *Cdkn1a* (*p21*) and decreased expression of the proliferation-associated gene *Ccnd1* (Fig 6B–6D). Moreover, EdU incorporation assays showed significantly reduced BAT cell proliferation upon *S100b* knockdown (Fig 6E and 6F), confirming S100B's essential role in supporting BAT cellular turnover. Functionally, this was accompanied by a significant drop in body temperature during TRF-STE, reversing the thermoprotective phenotype conferred by SCN lesioning (Figs 6G and S6A).

To assess sufficiency, we overexpressed *S100b* in BAT via Cre-dependent AAV-CMV-DIO-*S100b* co-injected with AAV-CAG-Cre-EGFP (Fig 6H). *S100b* overexpression downregulated *p21* and *Gadd45γ*, reduced p21 protein levels, and upregulated *Ccnd1* (Fig 6I–6K). These molecular changes translated into significantly increased BAT cell proliferation (Fig 6L and 6M) and resistance to TRF-STE-induced hypothermia (Figs 6N and S6B). Together, these data demonstrate that S100B is sufficient to drive preadipocyte proliferation and sustain thermogenesis under stress conditions.

As *S100b* expression in BAT is regulated by SCN activity, we next tested whether diverse modes of SCN disruption converge on similar outcomes. First, constant light exposure (LL), a noninvasive method known to induce behavioral arrhythmia and circadian disruption [37], was used as a physiological model of SCN impairment (Fig 7A). LL-exposed mice showed disrupted locomotor activity (S7A Fig) and a BAT molecular phenotype resembling SCN-lesioned mice, including elevated *S100b* and *Ccnd1*, and reduced p21 (Fig 7B–7D), enhanced EdU incorporation (Fig 7E and 7F), and attenuated TRF-STE-induced hypothermia (Figs 7G and S7B).

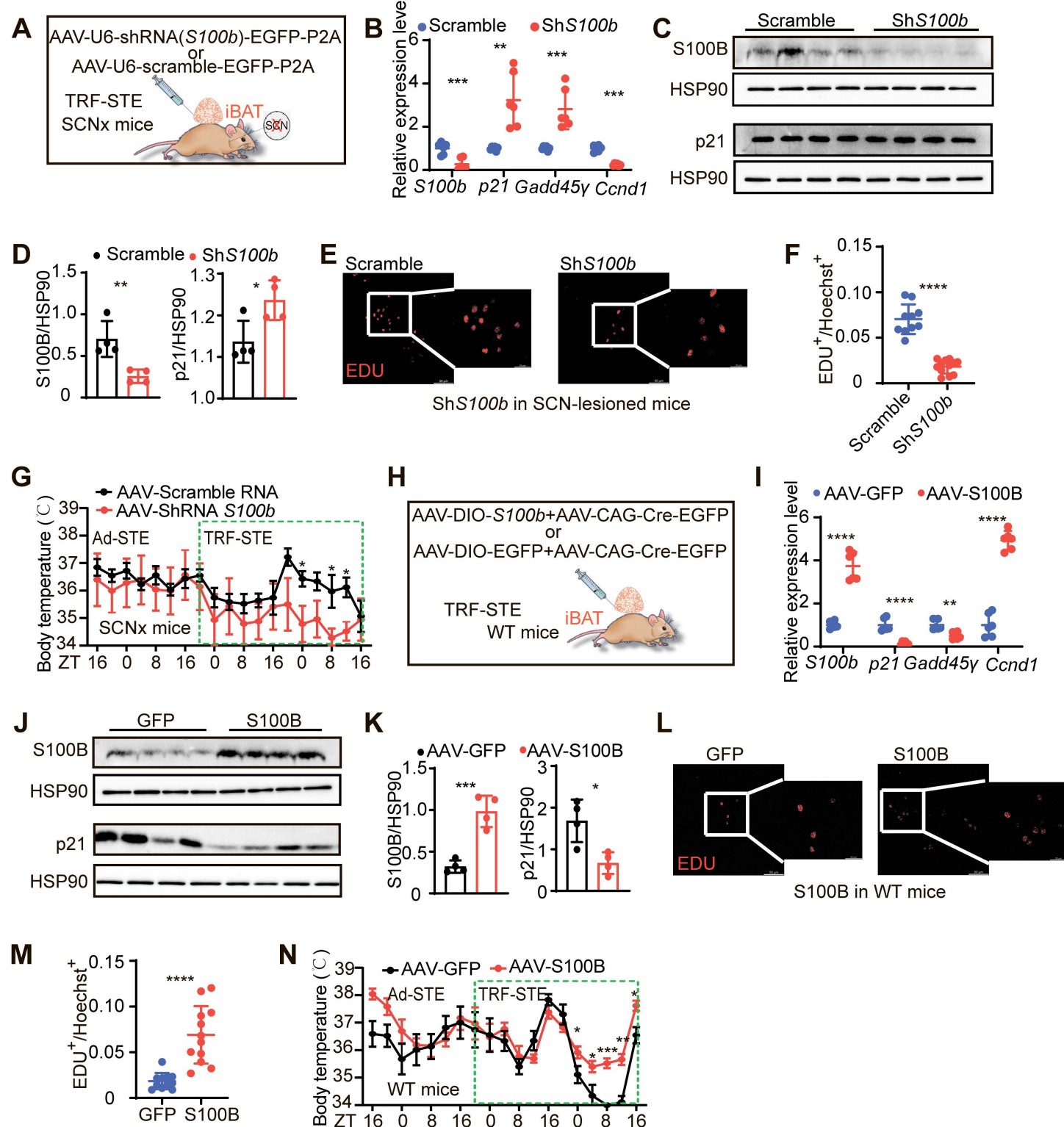

**Fig 6. SCN-mediated BAT thermogenesis via ADRB-S100B axis in response to TRF-STE. (A)** Schematic of AAV-mediated *S100b* knockdown or scramble control delivery into interscapular BAT of SCNx mice. **(B)** Relative mRNA expression of the indicated genes in interscapular BAT from SCNx mice under TRF-STE with scramble or *S100b* targeting shRNA. *n* = 6 per group. **(C and D)** Western blot analysis of S100B and p21 (**C**) and

corresponding densitometry quantification (**D**) in interscapular BAT from SCNx mice under TRF-STE ($n = 4$ per group). **(E and F)** EdU incorporation analysis in interscapular BAT from SCNx mice following *S100b* knockdown or scramble control. Representative images (**E**) and quantification of EdU-positive nuclei relative to total Hoechst-positive nuclei **(F)**. **(G)** Body temperature profiles of SCNx mice with *S100b* knockdown or scramble control under Ad-STE and TRF-STE (scramble: $n = 6$, *S100b* knockdown: $n = 5$). Data presented as mean ± SEM. Corresponding interscapular BAT and tail temperature data are shown in S6A Fig. **(H)** Schematic illustrating AAV-mediated overexpression of *S100b* (AAV-DIO-*S100b* + AAV-CAG-Cre) or EGFP control in interscapular BAT of WT mice. **(I)** Relative mRNA levels of indicated genes in interscapular BAT from EGFP or *S100b* overexpressing mice under TRF-STE ($n = 6$ per group). **(J and K)** Western blot analysis of S100B and p21 proteins (**J**) and corresponding densitometry (**K**) from interscapular BAT of EGFP or *S100b*-overexpressing mice ($n = 4$ per group). **(L and M)** EdU incorporation analysis in interscapular BAT from EGFP and *S100b*-overexpressing mice. Representative images (**L**) and quantification of EdU⁺ cells (**M**). $n = 6$ per group. **(N)** Body temperature responses under Ad-STE and TRF-STE in WT mice with EGFP or *S100b* overexpression in interscapular BAT. Data presented as mean ± SEM. $n = 6$ per group. See S6B Fig for interscapular BAT and tail temperature profiles. Unless otherwise indicated, data are presented as mean ± SD. $*p < 0.05$, $**p < 0.01$, $***p < 0.001$ and $****p < 0.0001$. Significance determined by unpaired two-tailed Student $t$ test (**B**, **D**, **F**, **I**, **K**, and **M**) or two-way ANOVA with Sidak'S multiple comparisons test (**G**, **N**). Scale bars, 50 µm **(E, L)**, 20 µm (magnified view of **E**, **L**). Schematic (**A**, **H**) created in BioRender.com. The data underlying the graphs shown in the figure can be found in S1 Source Data. Raw blot images can be found in S1 Raw Images.

To further confirm SCN regulation of this axis, we used two SCN-targeted lesion models: [1] NMDA-induced excitotoxicity, which causes focal damage to SCN neurons through glutamatergic overactivation (S7C Fig) [38]; and GABAergic neuronal ablation via AAV-DIO-Caspase3 delivered with AAV-VGAT1-Cre (S7D Fig). In both models, BAT exhibited elevated *S100b* and *Ccnd1* and suppressed *p21* expression, though to a lesser extent than in SCN lesioning, reinforcing that S100B activation is a conserved feature across diverse modes of SCN disruption (S7E and S7F Fig).

In summary, these findings establish that the SCN regulates BAT thermogenesis via a β3-adrenergic-S100B signaling axis. Across multiple independent models, SCN lesioning, light-induced arrhythmia, excitotoxicity, or targeted neuronal apoptosis, converge on S100B activation, promoting preadipocyte proliferation, suppressing senescence, and sustaining thermogenesis under thermal and nutritional stress. These results identify S100B as a critical downstream effector of SCN-driven sympathetic output, linking circadian control with metabolic resilience in adipose tissue.

## Discussion

Thermogenic adipose tissues, mainly BAT, are essential for maintaining energy homeostasis under conditions of thermal and nutritional stress. Here, we demonstrate that the SCN, the master circadian pacemaker, coordinates metabolic flexibility in BAT by integrating nutrient signals with sympathetic outflow to regulate substrate utilization and thermogenesis. Specifically, under TRF-STE, intact SCN activity suppresses BAT proliferation and promotes senescence, whereas SCN disruption elevates S100B expression mimicking HFD effect in the absence of lipids, promoting preadipocyte proliferation, and enables BAT to sustain thermogenesis through a glucose-driven program. Importantly, the positive feedback loop between S100B and ADRB3 sustains thermogenic output in the absence of lipid supply (Fig 7H).

Prior studies have established the importance of the SCN in coordinating metabolic and thermoregulatory responses during periods of energetic stress. For instance, SCN ablation impairs torpor induction during food deprivation [12,39], and TRF-STE induces severe hypothermia that is alleviated by SCN disruption or constant light exposure [14]. While the substrate flexibility of BAT in thermogenesis is well documented [40], the upstream neural mechanisms enabling this flexibility to nutrient availability under conditions of simultaneous thermal and nutritional stress have remained unclear.

Our results extend these findings by showing that SCN lesioning uncouples systemic EE from local BAT thermogenesis. While sham mice demonstrate metabolic flexibility by shifting from carbohydrate to lipid oxidation during TRF, SCN-lesioned mice fail to make this transition, instead maintaining a glucose-dependent profile. This loss of flexibility occurs despite reduced locomotor activity and lower systemic EE, indicating that SCN disruption reshapes energy partitioning at the tissue level. To test whether increased fuel availability alone could compensate for impaired central regulation, we evaluated HFD-fed sham mice. Despite significantly increased fat mass and thermoinsulation, these mice remained susceptible to TRF-STE-induced hypothermia, indicating that substrate abundance is insufficient without intact SCN signaling.

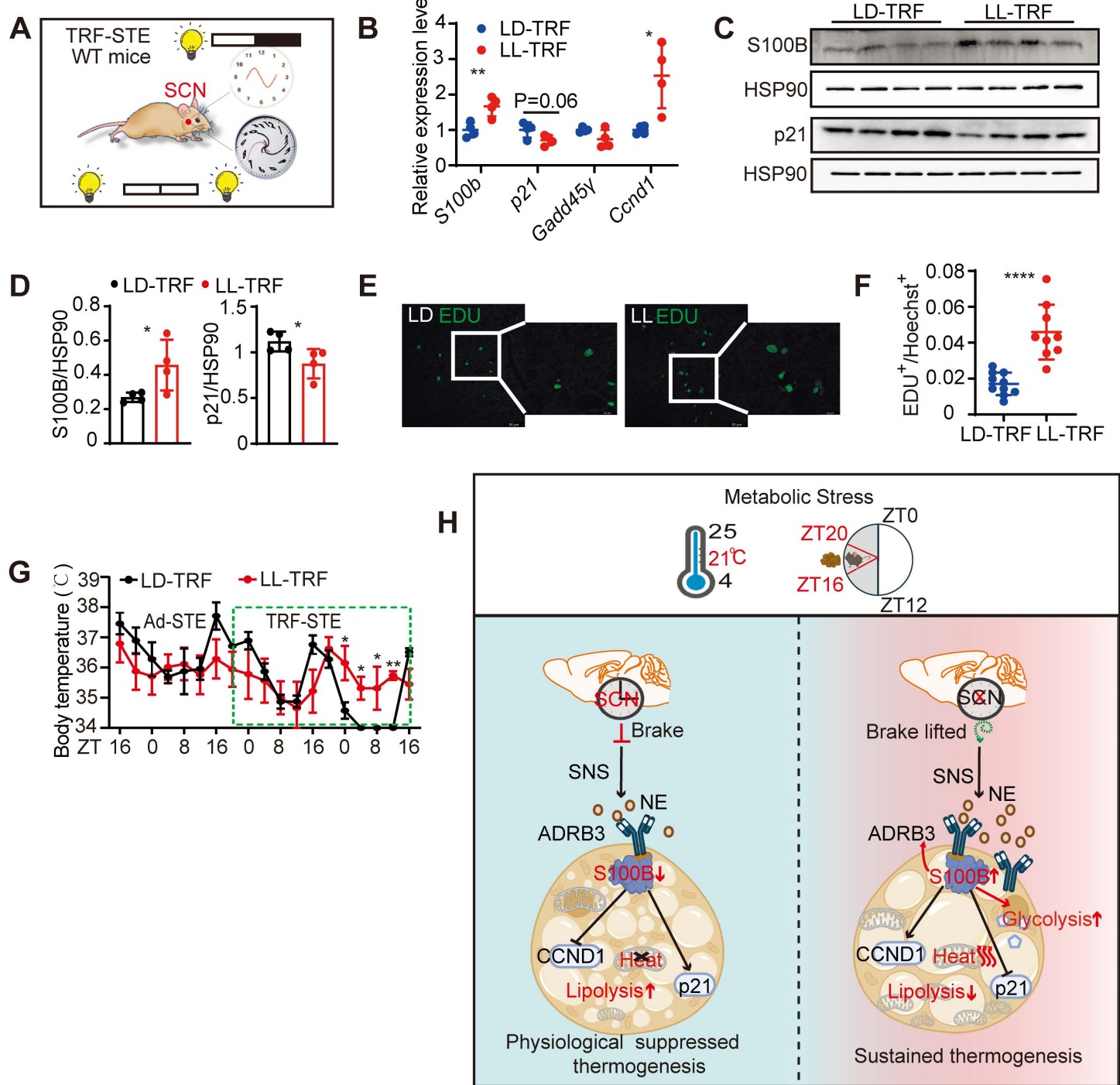

**Fig 7. Multiple complementary models of SCN dysfunction in response to TRF-STE. (A)** LL exposure paradigm is used to disrupt SCN rhythmicity. Arrhythmic mice were selected for subsequent analysis (S7A Fig). **(B)** Relative mRNA expression of indicated genes in interscapular BAT from WT mice under light/dark cycle (LD) or LL during TRF-STE ($n = 4$ per group). **(C and D)** Western blot analysis **(C)** and densitometry **(D)** of S100B and p21 proteins in interscapular BAT from mice under LD or LL conditions ($n = 4$ per group). **(E and F)** EdU staining in interscapular BAT from mice under LD or LL. Representative images **(E)** and quantification of EdU$^+$ cells **(F)** normalized to Hoechst staining. **(G)** Body temperature profiles of WT mice under LD or LL conditions during TRF-STE. Data presented as mean ± SEM. $n = 4$ per group. See S7B Fig for additional temperature metrics. **(H)** Working model illustrating the SCN-ADRB3-S100B axis in BAT during TRF conducted at ZT16-ZT20 in a subthermoneutral environment, the SCN regulates BAT thermogenic plasticity via SNS output. SCN lesioning enhances ADRB3 signaling and S100B expression, which together form a positive feedback loop that

amplifies β3-adrenergic sensitivity, promotes preadipocyte proliferation, suppresses senescence, and sustains glucose-driven thermogenesis. Unless otherwise indicated, data are presented as mean ± SD. *$p < 0.05$, **$p < 0.01$ and ****$p < 0.0001$. Significance determined by unpaired two-tailed Student $t$ test (**B**, **D**, and **F**) or two-way ANOVA with Sidak'S multiple comparisons test (**G**). Scale bars, 50 μm (**E**), 20 μm (magnified view of **E**). Schematic (**A**) and working model (**H**) created in BioRender.com. The data underlying the graphs shown in the figure can be found in S1 Source Data. Raw blot images can be found in S1 Raw Images.

This supports our central conclusion: effective thermogenic adaptation requires SCN-mediated sympathetic coordination, not merely energy reserves.

A significant contribution of this study is the identification of the ADRB3-S100B axis as a key effector of SCN-mediated BAT regulation. S100B expression is determined by nutrient availability under SCN control, being suppressed by fasting and thermoneutrality, and upregulated by HFD feeding; its functional impact on thermogenesis only becomes evident in the context of activation of β3-adrenergic signaling. We show that S100B physically interacts with ADRB3, increases receptor abundance, and enhances glycolytic activity in brown adipocytes, especially when combined with β3-adrenergic agonists. Although the mechanism by which S100B affects ADRB3 phosphorylation remains to be elucidated [35,36,41], their synergistic impact on ECAR suggests a sensitizing feedback loop that amplifies β3-adrenergic signaling. Unlike canonical thermogenic genes such as *Ucp1*, which respond to cold, *S100b* is nutrient-sensitive, downregulated by fasting and thermoneutrality, yet upregulated by HFD, highlighting its role as a metabolic amplifier attuned to energy supply. Thus, S100B acts not as a direct cold sensor but as a nutrient-sensitive amplifier of sympathetic input. This dual regulation reconciles our findings with previous reports that serum S100B is elevated in obesity and neuroinflammation. Whereas those studies reflected systemic or central changes, our data indicate that in BAT, S100B functions locally as an amplifier of β3-adrenergic signaling and a regulator of cellular plasticity.

S100B is a multifunctional calcium-binding protein implicated in stress responses, neuronal plasticity, and metabolic regulation [21,22]. It has been linked to BAT and beige adipocyte thermogenesis via enhanced sympathetic innervation [25,28,42]. Our study further demonstrates that S100B promotes the proliferation of BAT preadipocytes and represses senescence-associated genes. The observed increase in preadipocyte proliferation under SCN disruption likely represents an adaptive mechanism to maintain BAT thermogenic capacity during nutrient and thermal stress. Expanded pools of preadipocytes provide a sustained source of brown adipocytes, thereby enhancing tissue plasticity. Importantly, proliferative activity is coupled to a metabolic switch: while sham mice under TRF-STE suppress proliferation and favor lipid conservation, SCN-lesioned mice maintain proliferation alongside a shift toward glucose utilization. This coupling suggests that BAT expansion under SCN dysfunction is fueled by carbohydrate substrates, which provide a readily accessible energy source for proliferating cells and thermogenesis. Thus, proliferation not only sustains BAT mass but also integrates with substrate reallocation to preserve thermogenic output during energy stress, where lipid mobilization is impaired. These functions make S100B a compelling therapeutic candidate for augmenting EE and counteracting adaptive weight loss plateaus in dietary or pharmacological interventions [43,44].

Disruptions in SCN-regulated circadian rhythms, as seen in shift work, jet lag, and prolonged light exposure, impair SNS activity, weaken lipolysis, and disrupt energy homeostasis [7,9]. In this context, our findings hold strong physiological relevance, as they highlight how the SCN integrates central circadian timing with peripheral thermogenic regulation in response to combined environmental stressors, including nutrient scarcity and cold exposure. Although the TRF-STE paradigm is an experimental construct, it closely models ecologically relevant conditions encountered by nocturnal mammals during periods of seasonal food scarcity and low ambient temperatures. Thus, this model serves as a valuable tool for investigating the dynamic crosstalk between central circadian control and peripheral metabolic adaptation under conditions that reflect real-world physiological challenges.

Nonetheless, we acknowledge several limitations. First, although SCN lesioning is a well-established approach to abolish circadian output, it may introduce off-target effects unrelated to clock disruption. To address this, we employed multiple

orthogonal SCN perturbation models, such as constant light exposure, NMDA excitotoxicity, and caspase-3-mediated ablation. We observed consistent phenotypes across the models, which strengthened our conclusions. Second, our data are derived from male mice only; future studies are needed to determine the sex-specific generalizability of the ADRB3-S100B axis. Third, although we demonstrated a robust association between S100B expression and β3-adrenergic signaling, the molecular mechanisms by which S100B influences ADRB3 stability, trafficking, or signal transduction remain unclear. Future work employing high-resolution imaging and biochemical tools will be necessary to define this interaction in vivo. Moreover, while our study focuses on the SCN-BAT axis, thermoregulation is orchestrated by a broader hypothalamic network. Regions such as the preoptic area, arcuate nucleus, paraventricular nucleus, and dorsomedial hypothalamus, also participate in the integrative regulation of thermogenesis and metabolism [38,45–47]. Future studies are warranted to delineate their roles in concert with the SCN.

In summary, our study establishes the SCN as a central regulator of adaptive thermogenesis and an enabler of metabolic flexibility in response to environmental and nutritional stress. The identification of the ADRB3-S100B signaling axis provides a mechanistic framework for understanding SCN-driven BAT regulation and opens new avenues for therapeutic strategies targeting metabolic disorders associated with circadian disruption, such as obesity and insulin resistance.

## Materials and methods

| Reagent/resource | Reference or source | Identifier or catalog number |
|---|---|---|
| **Experimental models** | | |
| Mouse: C57BL/6J | GemPharmatech | Cat# N000013 |
| Cell lines: HEK293T | ATCC | Cat# CRL-3216 |
| **Recombinant DNA** | | |
| pCMV-S100B-eGFP | This study | N/A |
| pCMV-eGFP | This study | N/A |
| pCMV-ADRB3-mCherry | This study | N/A |
| pCMV-mCherry | This study | N/A |
| **Antibodies** | | |
| HSL | Cell Signaling Technology | Cat# 4107T |
| p-HSL (Ser660) | Cell Signaling Technology | Cat# 45804S |
| HSP90 | Proteintech | Cat# 13171-1-AP |
| ADRB3 | Bioss | Cat# BS-1063R |
| UCP1 | Proteintech | Cat# 23673-1-AP |
| PGC1α | Santa cruz | Cat# sc-13067 |
| p21 | Proteintech | Cat# 28248-1-AP |
| PDGFRα | Cell Signaling Technology | Cat# 3174S |
| S100B | ZEN-BIO | Cat# 380829 |
| mCherry | Proteintech | Cat# 26765-1-AP |
| EGFP | Proteintech | Cat# 66002-1-Ig |
| Total OXPHOS complex | Abcam | Cat# ab110413 |
| Alexa Fluor 647 goat anti-rabbit IgG | Thermo Fisher Scientific | Cat# A32733 |
| Anti-rabbit IgG HRP-linked | Cell Signaling Technology | Cat# 7074s |
| Anti-mouse IgG HRP-linked | Cell Signaling Technology | Cat# 7076s |
| TH (WB) | Proteintech | Cat# 25859-1-AP |
| GFAP | Dako | Cat# Z0334 |
| TH (Immunofluorescence) | Abcam | Cat# ab152 |
| CCND1 | Proteintech | Cat# 26939-1-AP |

| Reagent/resource | Reference or source | Identifier or catalog number |
|---|---|---|
| **Oligonucleotides and other sequence-based reagents** | | |
| Primers for RT-PCR | This study | S5 Table |
| Scramble shRNA sequence: CCTAAGGTTAAGTCGCCCTCG | BrainVTA | N/A |
| *S100b* shRNA sequence: GCACAAGCTGAAGAAGTCAGA | BrainVTA | N/A |
| **Chemicals, Enzymes, and other reagents** | | |
| DMEM medium/ High Glucose | Gibco | Cat# C11995500BT |
| Fetal bovine serum | Gibco | Cat# 10,270 |
| Penicillin-streptomycin | Gibco | Cat# 15070 |
| LipoMax | SUDGEN | Cat# 32012 |
| Isoflurane | RWD | Cat# 845AS |
| Ethanol | SINOPHARM | Cat# 10009259 |
| Trizol Reagent | Ambion | Cat# 15596018 |
| Chloroform | Sigma | Cat# 288306 |
| Isopropanol | SINOPHARM | Cat# 80109218 |
| DEPC | BBI | Cat# B600154 |
| 5-ethynyl-2′-deoxyuridine | Sigma | Cat# 900584 |
| Optimal Cutting temperature | SAKURA | Cat# 4583 |
| BSA | Sangon | Cst# A500023 |
| DAPI | Sigma | Cat# D9542 |
| Recombinant Mouse S100B | Targetmol | Cat# TMPJ-00990 |
| SR59230A | MCE | Cat# HY-100672 |
| NMDA | MCE | Cat# HY-17551 |
| Cycloheximide | MCE | Cat# HY-12320 |
| Collagenase type I | Sigma | Cat# SCR103 |
| Trypsin | Biosharp | Cat# BL501A |
| Indomethacin | Selleck | Cat# S1723 |
| Dexamethasone | Sigma | Cat# 265005 |
| Rosiglitazone | Sigma | Cat# R2408 |
| Insulin | Procell | Cat# PB180432 |
| Glucose | Sigma | Cat# G8270 |
| FCCP | Sigma | Cat# C2920 |
| Oligomycin | Sigma | Cat# O4876 |
| Antimycin A | Sigma | Cat# A8674 |
| Rotenone | Sigma | Cat# R8875 |
| ADP | Targetmol | Cat# T1723 |
| GDP | Targetmol | Cat# T7210 |
| Sodium pyruvate | Sigma | Cat# P2256 |
| Malic acid | Sigma | Cat# M8304 |
| Isobutylmethylxanthine | Sigma | Cat# I5879 |
| **Software** | | |
| Clocklab | Actimetrics, Evanston | https://actimetrics.com/products/clocklab/ |
| GraphPad Prism 8 | GraphPad Software, San Diego, CA, USA | https://www.graph-pad-prism.cn/ |

| Reagent/resource | Reference or source | Identifier or catalog number |
|---|---|---|
| ImageJ | National Institutes of Health, USA | https://imagej.net/ij/ |
| IBM SPSS Statistics | International Business Machines, USA | https://www.ibm.com/spss |
| LAS X | Leica microsystems | https://www.leica-microsystems.com/ |
| Flir | Tools Flir | https://flir.custhelp.com |
| **Other** | | |
| Virus strains | | |
| PRV-CAG-EGFP | BrainVTA | Cat# P01001 |
| HSV-EGFP | BrainVTA | Cat# H01001 |
| AAV2/8-U6-sh*S100b*-EGFP | BrainVTA | Cat# PT1019 |
| AAV2/8-U6-scrambleRNA-EGFP | BrainVTA | Cat# PT0900 |
| AAV2/8-CMV-DIO-*S100b*-EGFP | Obiosh | Cat# H27455 |
| AAV2/8-CMV-DIO-EGFP | Obiosh | Cat# H5010 |
| AAV2/8-CAG-Cre-EGFP | Obiosh | Cat# CN390 |
| AAV2/9-EF1a-DIO-taCasp3-EGFP | BrainVTA | Cat# PT3345 |
| AAV2/9-EF1a-DIO-EGFP | BrainVTA | Cat# PT0795 |
| AAV2/9-VGAT1-CRE-mCherry | BrainVTA | Cat# PT0533 |
| Critical commercial assays | | |
| S100B ELISA | LunchChangShuo Biotech | Cat# ED-20090 |
| SYBR Green Mix/ROX qPCR | Takara | Cat# RR420 |
| NA ELISA | Elabscience | Cat# E-EL-0047 |
| Click-iT Edu Imaging Kit 488 | Invitrogen | Cat# C10337 |
| Click-iT Edu Imaging Kit 647 | Invitrogen | Cat# C10340 |
| Senescence β-Gal Staining Kit | Beyotime | Cat# C0602 |
| DAB solution mixture | MXB biotechnologies | Cat# KIT-5920 |
| FastPure DNA Isolation Kit | Vazyme | Cat# DC112 |
| Seahorse XF Glycolysis Stress Test Kit | Agilent Technologies | Cat# 103020 |
| Non-shape-stable soft tissue NSIII Tissue Clearing Kit | Nuohai Life Science | Cat# 240920-12 |

## Animal and ethics statement

All animal experiments were conducted in accordance with *the Regulation for the Administration of Affairs Concerning Experimental Animals of the People's Republic of China and the national standard Laboratory Animals: Guideline for Ethical Review of Animal Welfare* (GB/T 35892-2018). All procedures were reviewed and approved by the Animal Care and Use Committee of Soochow University under protocol numbers YX-19-3 and YX-24-02.

C57BL/6J mice were purchased from GemPharmatech Co. (Nanjing, China) and maintained in a pathogen-free (SPF) animal facilities under a 12-hour light/12-hour dark cycle at controlled temperature and humidity. Animals had ad libitum access to food and water unless subjected to TRF protocols as indicated. To induce obesity, 8-week-old C57BL/6J male mice were fed a HFD containing 60% fat, 20% carbohydrate, and 20% protein (#XTHF60, Xietong, China) for 7 weeks. Body weight and body fat composition were monitored and recorded.

## Cell line

HEK293T cells are cultured in a high-glucose DMEM medium containing 10% fetal bovine serum and penicillin-streptomycin. Depending on the experimental requirements, cells are cultured in 10 cm, 35 mm, or 4-chamber

coverslips. HEK293T cells are transfected with plasmids using LipoMax Transfection (#32012, SUDGEN, China) according to the manufacturer's instructions. Observe fluorescence and capture under a confocal fluorescence microscope (TCS SP8, Leica) 24 h after transfection. Alternatively, cells can be collected for coIP or protein stability assay.

**SCN-lesioned mice**

Eight-week-old male C57BL/6J mice were anesthetized with isoflurane and maintained on a heating pad to preserve CBT throughout the procedure. After fixation in a stereotaxic apparatus, a 0.5 mm diameter burr hole was drilled into the skull, and a 0.15 mm diameter platinum-iridium electrode (custom-fabricated by Suzhou Kedo Brain Technology Co.) was stereotactically lowered into the SCN. The electrode was polyimide-coated except for the tip. The stereotaxic coordinates used to target the SCN were: anteroposterior (AP): −0.05 mm, mediolateral (ML): ±0.15 mm, dorsoventral (DV): −5.85 mm from the bregma. A lesion was induced by passing a 0.4 mA direct current for 30 s using a Ugo Basile lesioning device (Model 53500). Sham-operated mice underwent identical procedures except that electrodes were inserted only to 5.4 mm DV depth, and no current was applied. After 1 week of postoperative recovery, all mice were placed in wheel-running cages under a 12-hour light/12-hour dark (LD) cycle for 1 week to assess circadian locomotor activity. Periodogram analysis showed that all sham mice retained robust circadian rhythms, whereas approximately 30% of SCNx mice became arrhythmic (S1A Fig). For downstream physiological and molecular experiments, only SCN-lesioned mice that completely lost circadian rhythmicity and had comparable average body weight and size to sham controls were included. At the conclusion of the experiments, histological analysis was performed to verify the accuracy and extent of SCN lesions [14]. For excitotoxic lesion experiments, 30 nl of NMDA (50 mM) was bilaterally injected into the SCN using the same stereotaxic coordinates. Two weeks later, wheel-running assays were performed to confirm circadian disruption, followed by GFAP immunostaining to assess gliosis and lesion specificity. For the viral-mediated lesioning, AAV2/9-DIO-Casp3 ($5.49 \times 10^{12}$ vg/ml) and AAV2/9-VGAT1-Cre ($3.2 \times 10^{12}$ vg/ml) were mixed in a ratio of 2:1, and a total of 150 nl was bilaterally injected into the SCN. The control group received AAV2/9-DIO-EGFP ($5.21 \times 10^{12}$ vg/ml) co-injected with AAV2/9-VGAT1-Cre. Behavioral circadian rhythm was assessed via wheel-running activity 2 weeks postinjection.

**Mice locomotor analyses**

After surgery, the mice were housed individually in cages with running wheels, had free access to food and water, and were maintained under a 12-hour light-dark cycle. The locomotor rhythms were recorded using Clocklab (V6.1.02) with continuous recording for 1 week. Arrhythmic mice were selected for subsequent manipulations.

**Constant light treatment mice**

Eight-week-old male mice were individually housed in boxes that could regulate the light environment. The experimental group of mice was exposed to constant light, while the control group was maintained on a 12-hour light-dark cycle, with a light intensity of 200 lux for both groups. This treatment lasted for at least 2 months, and the mice's free wheel-running activity was recorded.

**Different temperatures and nutritional conditions**

Eight-week-old male mice were individually housed in an incubator (PGX-350B, Ningbo Saifu) with adjustable temperatures. After a one-week acclimation period at room temperature, they were placed under different temperature conditions or given various dietary patterns (chow diet, ZT16-20 TRF, fasting, or HFD). After a 48-hour treatment, the mice were euthanized, and their interscapular BAT was rapidly collected.

## TRF-STE

SCN-lesioned mice and sham mice were individually housed in cages placed in an incubator with adjustable temperature and light. The temperature control of the incubator is based on a high-precision temperature control module, ensuring variations of less than 1°C. Each layer of the incubator is checked with a mercury-in-glass thermometer to ensure that the temperature of each layer is uniform. The light schedule is 12 h of light (light intensity is approximately 200 lux) followed by 12 h of darkness. The mice were first given *ad libitum* at 21°C for 2 weeks to acclimate to the incubator. Then they were then placed in new, clean cages, and the food was removed at ZT20. Each cage was covered with wood shavings to provide a small nest. The mice were given food on the following day (ZT16), which was the first day of the TRF-STE protocol. Calculate and record the amount of food eaten by the mice each day.

## Infrared thermograph and core body temperature recording

The backs of the mice were shaved 2 days before recording, and a Thermochip (SureFlap Ltd) was injected into the abdominal cavity of the mice to detect CBT. Changes in CBT of mice were detected using a thermochip monitoring device (GPR+, Destron Fearing). An infrared imager (Flir E8) was used to detect the surface temperature of the BAT of the mice from a top view. The camera was kept at the same height for each detection, and the images were automatically synthesized by the device and processed using Flir Tools software.

## Indirect calorimetry

Mice were individually acclimated for 3 days in metabolic cages of a comprehensive animal monitoring system (Oxymax, Columbus Instruments) and provided with food and water ad libitum. The room lighting followed a 12-hour light/12-hour dark cycle, and the temperature was freely adjusted to meet the different experimental needs. For TRF mice, food was provided 4 h after the lights were turned off (ZT16), and food was removed 8 h after the lights were turned off (ZT20). $O_2$, $CO_2$, RER ($CO_2/O_2$), EE, and locomotor activity were recorded throughout the experiment.

## Interscapular BAT adeno-associated virus (AAV) injection

After being anesthetized with isoflurane, the fur over the scapular region of SCN-lesioned or 8-week-old male C57BL/6J mice was shaved, and the skin was disinfected with 75% alcohol. An incision was made to expose the interscapular BAT pads bilaterally, and a total of $1 \times 10^{12}$ vg/ml AAV was injected into multiple sites of each pad using a microsyringe. The mice were then allowed to recover for 1 week before entering the TRF-STE protocol. The following viral vectors were used: AAV2/8-U6-sh*S100b*-EGFP ($5.08 \times 10^{12}$ vg/ml) and AAV2/8-U6-scramble RNA-EGFP ($5.08 \times 10^{12}$ vg/ml), which were purchased from BrainVTA (China); and AAV2/8-CMV-DIO-*S100b*-EGFP ($4.3 \times 10^{12}$ vg/ml) and AAV2/8-CMV-DIO-EGFP ($4.5 \times 10^{12}$ vg/ml), which were obtained from Obiosh (China).

## Anterograde, retrograde tracing, and virus

The retrograde tracing viral tracer used in this study was the pseudorabies virus (PRV-CAG-EGFP). After the 8-week-old male C57BL/6J mice were anesthetized with isoflurane, an incision was made in the skin at the site of the interscapular BAT to expose the fat pads on both sides. A total of 3 μl of the virus was injected into both sides of the interscapular BAT at multiple points using a microsyringe (Size: 2 μl, RWD). The Brain slices were removed for observation on days 3, 4, and 5.

The anterograde virus used in this study was the herpes simplex virus (HSV-EGFP). After the mice were anesthetized, 200 nl of HSV was administered to the unilateral SCN. After seven days, the interscapular BAT was removed, fixed, sliced, and observed.

## Mouse PET scans and gamma counting

PET/CT (SNPC-304, Pingseng Technology) scans, and gamma counting (WIZARD 2,480, PerkinElmer) mouse scans were performed at the First Affiliated Hospital of Soochow University. On the second day of TRF-STE, the mice were transported using an onboard refrigerator to facilitate temperature control of the environment. After the mice were anesthetized with isoflurane, $^{18}$F-FDG was injected into the tail vein. Sixty minutes later, the mice were scanned with PET/CT, and the mice were always anesthetized. At the end of the scan, the mice were killed by cervical dislocation, and organs were rapidly harvested and weighed for the gamma counter to quantify radioactivity, which was normalized based on organ weight and dose at the time of the scan. The percentage of the injected dose per gram was calculated for the tissues.

## SnRNA-seq

SnRNA-seq and analysis of interscapular BAT were performed by Shanghai OB Biotech Co., at ZT16 on the second day of TRF-STE. Mice were euthanized, and interscapular BAT was rapidly dissected. Each group of samples consists of interscapular BAT from five mice. These tissues were processed into a cell suspension. The cell suspension was filtered twice through a 40 µm cell strainer (#352340, BD), centrifuged at 300 rpm for 5 min at 4°C, and the pellet was resuspended after the addition of red blood cell lysis buffer (#130-094-183, MACS). After standing for 10 min, centrifuge at 300 rpm for another 5 min, remove the supernatant, resuspend the cell pellet, and count the cells (Luna, LUNA-II). The cell concentration was then adjusted to 1,000 cells/µl.

Single-cell libraries were prepared for all samples using 10× Genomics Chromium Next GEM Single Cell 3′ Reagent Kits V3.1 (1000268, Chromium). The well-constructed library was subjected to high-throughput sequencing on the Illumina NovaSeq 6000 PE150 platform. Based on a preliminary quality control using Cell Ranger (V7.1.0), further quality control of experimental data was performed to exclude data from multiples, doublets, or unbound cells. This is followed by downstream analysis. In this project, we used PCA (Principal Components Analysis) and UMAP (Uniform Manifold Approximation and Projection) algorithms for dimensionality reduction. Based on the results of PCA, UMAP was used to visualize the clustering of single-cell populations. The clustering algorithm used was SNN (Shared Nearest Neighbors), which ultimately resulted in the optimal cell grouping. The Presto test method is used to perform differential expression analysis between the specified cell population and all other cell populations. Filtering criteria include a log fold change > 0 and a minimum percentage (min.pct) of gene expression across all cells > 0.25. This process helps to identify marker genes for each cell population. The top 10 marker genes are ranked from highest to lowest based on Gene Diff values. Pct1 represents the proportion of cells expressing the marker gene within the current group of cells. Pct2 represents the proportion of cells expressing the marker gene in the remaining groups. The ratio of these two percentages is the value of gene-diff (S1 Table).

$$Gene\ Diff = \frac{pct1}{pct2}$$

## RNA-seq

At ZT4 and ZT16 on the second day of TRF-STE, mice were euthanized by cervical dislocation, and interscapular BAT was collected and stored in liquid nitrogen. Total RNA from interscapular BAT was extracted using Trizol reagent (Ambion) and chloroform, and then the concentration was detected using NanoDrop 2000 (Thermo Fisher). RNA-seq was performed by Genewiz (Suzhou, China). First, the total RNA was quality-controlled using Agilent 2200 tapstation, then RNA-seq was performed on an Illumina NovaSeq 6000 platform with PE 150-bp reads, and sequencing was performed using Illumina bcl2fastq. Raw binary base calling data were converted to sequence data. Sequencing data were compared to

the mm10 reference genome at UCSC using Bowtie (V1.1.2). Finally, FPKM values of the genes in each sample were calculated using RSEM (V1.2.8). Screening for differentially expressed genes was based on fold change > 2, $p < 0.01$.

### Ingenuity Pathway Analysis (IPA)

Pathway enrichment for the differentially expressed genes from RNA-seq was analyzed by IPA (PMID: 24336805) ($p < 0.01$). The highest-ranking canonical pathways were selected, focusing on those with significant activation z-scores at both ZT4 and ZT16 (absolute [Z score] < −2). The data is presented in S3 Table.

### The Alphafold2 model of the ADRB3/S100B complex

Five models of the ADRB3/S100B complex were generated by ColabFold 1.5.5 in the local computer (https://github.com/YoshitakaMo/localcolabfold) using human ADRB3 (Uniprot ID P13945) and S100B (Uniprot ID P04271) sequences, and both pdb templates and amber refinement were used during modeling. The top-one-ranked model was chosen for the structure analysis using PyMOL (https://pymol.org/).

### EDU labeling in vivo

On the day before TRF-STE, mice were injected intraperitoneally with EdU (25 μg/g body weight, #900584, Sigma), with injections every 8 h for a continuous period of 3 days and a final injection 2 h before euthanasia. The mouse interscapular BAT was collected and preserved in 4% PFA. This was then cut into 7 μm sections (RM2106, Leica) and stained using the Click-iT Edu Imaging Kit (C10340-red or C10337-green, Invitrogen). After washing with PBS, the sections were immersed in a DAPI staining solution (1:5000 in PBS) for 10 min at room temperature. After the slides were sealed with 50% glycerol (Sangon, China), they were observed and photographed using an inverted fluorescence microscope (IX53, OLYMPUS) or a confocal microscope (TCS SP8, Leica), and the EdU-positive cells were counted within the same field of view, and the ratio to the total number of cells was calculated.

### Protein stability assay

Transfect HEK293T cells with the expression plasmid. After 24 h of transfection, treat the cells with the protein synthesis inhibitor cycloheximide (CHX, MCE) at a concentration of 20 μg/ml. The cell suspension is then collected at the appropriate time and subjected to western blot analysis.

### Stromal vascular fraction (SVF) cell culture

Brown adipose tissue SVF was isolated from 6-week-old C57BL/6J male mice. Mice were killed by cervical dislocation, and the interscapular BAT was dissected out and rinsed three times with pre-cooled PBS at 4°C. After mincing, the tissue was treated with type I collagenase (SCR103, Sigma) at 37°C for 40 min. After centrifugation at 1,200$g$ for 2 s, the upper layer of the cell suspension was collected, followed by centrifugation at 1,700$g$ for 5 min to collect the lower layer of pelleted cells. The cells are then resuspended in a high-glucose DMEM medium (containing 20% fetal bovine serum and penicillin-streptomycin), plated into 35 mm dishes, and cultured at 37°C with 5% $CO_2$.

For EdU-labeled cell proliferation of SVF, 0.1 μM recombinant S100B protein (TMPJ-00990, Targetmol USA) or saline and 10 μM EdU were added to the medium. After 48 h, cells were washed three times with PBS, fixed with 4%PFA for 10 min, and visualized using the Click-iT Edu Imaging Kit. The nucleus was labeled with DAPI for 10 min at room temperature.

For SVF immunofluorescence, after the cells were fixed with 4%PFA, they were blocked with 5% normal donkey serum for 1 h at room temperature. Anti-PDGFRα antibody (1:500, Cell Signaling Technology, 3174S) was then added and incubated overnight at 4°C. The next day, after washing three times with PBS, they were incubated with the secondary

antibody (goat anti-rabbit, Alexa Fluor 647) for 1 h at room temperature in the dark. After nuclear labeling with DAPI, the cells were imaged by confocal fluorescence microscopy.

For the analysis of cell growth curves, after the addition of 0.1 μM S100B recombinant protein, the cell culture plate was placed in a fully automated live cell dynamic imaging system (Incucyte SX5, Sartorius), and the cell count was recorded every hour for a continuous period of 72 h. Seahorse extracellular flux analysis for glycolytic stress test.

SVF was isolated from the interscapular BAT of 6-week-old C57BL/6J male mice. SVF was added to Seahorse XF24 cell culture microplates to induce differentiation. Change the growth medium every other day until cells reach 80%−100% confluence. The growth medium was replaced with an induction medium containing 0.64 μM dexamethasone, 0.625 mM isobutylmethylxanthine, 8.4 μM rosiglitazone, and 1.6 μg/ml insulin. The induction medium was changed every other day and replaced with a differentiation medium (composition is the same as the induction medium except for the addition of 1.63 μM 3, 3′, 5-triodo-L-thyronine sodium salt). The differentiation medium was changed every other day, and the glycolytic stress test was performed. Glycolysis stress test: The Agilent Seahorse XF Glycolysis Stress Test Kit was used according to the standard Agilent Seahorse protocol. Prior to the glycolytic stress test, the SVF medium was pretreated with 0.1 μM/0.2 μM recombinant S100B protein and/or 0.1 μM ADRB3 agonist CL316243 for 18 h. After six consecutive measurements of basal ECAR, 10 mM glucose, 2 μM oligomycin, and 50 mM 2-DG were sequentially injected for three ECAR measurements. Wave (Agilent Technologies V2.6.4) software was used to analyze the parameters.

### SVF induction differentiation

SVF was isolated from the interscapular BAT of 6-week-old C57BL/6J male mice. First, SVF was cultured using a growth medium. After the cells became fully confluent (24 h later), the growth medium was replaced with an induction medium. The experimental group was additionally supplemented with 0.1 μM recombinant S100B protein, while the control group received only fresh medium. The differentiation medium was refreshed every 2 days. Cells were collected on days 0, 3, and 5 of induction. RNA was extracted from the collected cells, reverse transcribed, and used for qPCR to detect the expression of adipocyte differentiation-related genes.

### Measurement of mitochondrial oxidation

The OCR was measured using a microfluorimetric Seahorse XF24 Analyzer (Agilent Technologies). Mice that had undergone 2 days of TRF were sacrificed, and their interscapular BAT was quickly removed. Mitochondria were isolated from the tissue using MIB (250 mM sucrose, 5 mM HEPES, 2 mM EGTA, and 1% BAS, PH = 7.2) as follows: the tissue was homogenized in MIB, then centrifuged at 800$g$ to remove debris, and finally centrifuged at 10,000$g$ to collect the mitochondria. The mitochondrial pellet was resuspended in MAS (155 mM KCl, 10 mM $KH_2PO_4$, 2 mM $MgCl_2$, 5 mM HEPES, 1 mM EGTA, plus 0.1% BSA). The assay solution also contained 10 mM pyruvate, 5 mM malate, and 1 mM GDP, with the pH adjusted to 7.2. The assay protocol was as follows: Port A: 5 mM ADP; Port B: 4 μM oligomycin; Port C: 5 μM FCCP; Port D: 5 μM antimycin A and 5 μM rotenone. Wave (Agilent Technologies V2.6.4) software was used to analyze the parameters.

### Seahorse analysis of ex vivo BAT

On day 2 of TRF-STE, at ZT16, mice were euthanized, and the interscapular BAT was quickly removed, rinsed with PBS, weighed at 5 mg, and placed into the Seahorse XFe24 Islet Capture FluxPak, and then minced. Each well was filled with 500 μl of XF DMEM Medium (with 2 mM glutamine) and incubated at 37°C for 50 min. The tissue's ECAR was detected using the microfluorimetric Seahorse XF24 Analyzer, with 10 mm glucose added during the detection. Wave (Agilent Technologies V2.6.4) software was used to analyze the parameters.

## Whole-mount TH immunostaining and 3D imaging

On the second day of TRF-STE at ZT16, mice were anesthetized with pentobarbital sodium and slowly perfused with PBS, followed by 4% PFA. Interscapular BAT was excised, fixed in 4% PFA at 4°C overnight, and washed twice with PBS the next day. The fixed tissue was cleared using the NSIII Tissue Clearing Kit (#240920-12, Nuohai Life Science, Shanghai) according to the manufacturer's instructions. Cleared samples were incubated in primary antibody solution (Abcam, AB152) at 37°C with gentle shaking for 15 days, washed with PBS, and then immersed in secondary antibody solution (Thermo Fisher Scientific, A32733) at 37°C for 10 days. After refractive-index matching, 3D fluorescence imaging was performed on a Nuohai LS-18 tiling light-sheet microscope, and images were rendered in Amira (Thermo Fisher Scientific). Nerve-fiber signals were segmented and traced using the segmentation and filament modules of Amira, and final data were visualized with Amira's rendering tools.

## Immunohistochemistry for interscapular BAT sections

After the mice were anaesthetized by intraperitoneal injection of sodium pentobarbital (4% in saline), the mice were perfused with PBS and 4% PFA. Mouse interscapular BAT was collected and processed into 7 μm paraffin sections. The sections were then deparaffinized and antigen retrieved with a sodium citrate antigen retrieval solution (pH 6.0). After washing with PBS, endogenous peroxidase was removed with 3% $H_2O_2$, the sections were incubated with 0.3% TritonX-100 (Sigma) for 20 min at room temperature, washed in PBS and incubated with 10% (in PBS) normal goat serum for 1 h at room temperature, and the primary antibody rabbit anti-S100B (1:200,380829, ZEN-BIO) or PDGFRα (1:500, Cell Signaling Technology, 3174S) was diluted in blocking solution and incubated overnight at 4°C. The next day, the primary antibody was washed off with PBS and incubated with horseradish peroxidase-conjugated secondary antibody for 1 h. The final sections were stained with a DAB staining solution mixture (KIT-5920, MXB biotechnologies China) for visualization, and hematoxylin was used to label the nucleus.

## Western blotting and quantification

Protease and phosphatase inhibitors (Roche) were added to the RIPA lysis buffer. The mouse interscapular BAT was homogenized thoroughly in the RIPA buffer, then centrifuged at 12,000$g$ for 15 min at 4°C, and the supernatant was collected. The protein was quantified using the bicinchoninic acid assay (P0011, Beyotime China), and the total protein was diluted to a concentration of 1 μg/μl. SDS-PAGE gels of different concentrations were used to quantify the corresponding proteins. The dilution ratios for the primary antibodies were as follows: HSP90 (1:1000, Proteintech, 13171-1-AP), UCP1 (1:1000, Proteintech, 23673-1-AP), HSL (1:1000, Cell Signaling Technology, 4107T), p-HSL (Ser660) (1:1000, Cell Signaling Technology, 45804S), Total OXPHOS complex (6 μg/ml, Abcam, ab110413), PGC1α (1:1000, Santa Cruz, sc-13067), S100B (1:500, 380829, ZEN-BIO), p21 (1:1000, Proteintech, 28248-1-AP), ADRB3 (1:1000, Bioss, bs-1063R), CCND1 (1:5000, Proteintech, 26939-1-AP). Immerse the PVDF membrane (Millipore) in the diluted primary antibodies and incubate overnight. After three washes in TBST, the membrane was incubated with peroxidase-conjugated secondary antibodies (anti-rabbit/mouse IgG H + L HRP-linked, 7074s/7076s, Cell Signaling Technology) for 1 h at room temperature. Finally, the membranes were imaged using an automatic chemiluminescence imaging system (ChemiScope S6, Clinx Shanghai) and quantified using ImageJ (National Institutes of Health).

## Quantitative RT-PCR

The total RNA was measured using the NanoDrop 2000 (Thermo Fisher). The PrimeScript RT Reagent Kit (RR037A, Takara) was used for genomic DNA isolation and reverse transcription to obtain cDNA. To quantify mDNA, a DNA Isolation Kit (DC102, Vazyme) was used to extract the genomic DNA from interscapular BAT according to the manufacturer's instructions, and quantitative PCR was performed using SYBR Green (RR420, Takara) with the ViiA7 (Life Technologies).

The relative expression levels of genes were calculated using the $2^{\triangle\triangle CT}$ (cycle threshold) method, and the final results were normalized to the expression level of actin.

### ELISA

Mice were killed by cervical dislocation, fresh blood was collected, centrifuged at $3000g$ for 10 min at room temperature, and the serum was collected and stored in an ultra-low temperature freezer. The mouse S100B ELISA kit (#ED-20090, Lunch Chang Shuo Biotech, China) or NE ELISA kit (#E-EL-0047, Elabscience) was used according to the manufacturer's instructions to detect the level of S100B or NE in each serum sample.

### Denervation of the sympathetic nervous system

6-OHDA (10 mg/ml, #H4381, Sigma) was dissolved in 0.15 mol/l saline containing 1% ascorbic acid. A midline skin incision was then made along the upper dorsal surface to expose both interscapular BAT pads, and 10 µg of 6-OHDA was injected into each pad using a 10 µl microsyringe. Perform TRF-STE one week after injection.

### Quantification and statistical analysis

All experimental data were first tested for normal distribution using the Shapiro–Wilk test in GraphPad Prism V8.0.2 (GraphPad, USA). Subsequently, Student $t$ tests or two-way ANOVAs were performed based on experimental needs, while GLM for mouse EE was calculated using SPSS Statistics V25.0 (IBM, USA), with mouse body weight as the covariate and EE as the dependent variable. Statistical details of the experiments were provided in the legends and corresponding methods sections. $P$-values are indicated in the Figs. Statistical significance is indicated as $p < 0.05$ (*$p < 0.05$, **$p < 0.01$, ***$p < 0.001$, and ****$p < 0.0001$), NS: not significant.

### Supporting information

**S1 Fig. Energy metabolism of SCN-lesioned, sham mice.** (**A**) Representative actograms showing locomotor activity patterns in sham (top) and SCNx (bottom) mice used for screening of behavioral arrhythmicity following SCN lesioning. (**B**) CBT profiles in WT mice subjected to TRF-STE initiated at different circadian windows (ZT4-8, ZT10-14, and ZT22-2). Arrows indicate the time of TRF onset. Data are reanalyzed from Zhang and colleagues, 2020 and presented as mean ± SEM ($n = 4$ per group). (**C**) Representative 24-hour plot of EE measured by indirect calorimetry in sham and SCNx mice under Ad-STE. Data shown as mean ± SEM (sham: $n = 11$; SCNx: $n = 10$). (**D–F**) Total $O_2$ consumption (**D**), $CO_2$ production (**E**), and EE (**F**) during Ad-STE. Data presented as mean ± SEM (sham: $n = 11$; SCNx: $n = 10$). (**G**) Three-day EE profiles under TRF-STE measured by indirect calorimetry. Data presented as mean ± SEM ($n = 10$ per group). (**H**) GLM analysis of EE under TRF-STE, using group (sham versus SCNx) as the independent variable and body weight as a covariate. Sham: $n = 11$; SCNx: $n = 10$. (**I** and **J**) Statistical analysis of total physical activity from the second day under TRF-STE (**I**, $n = 7$) and Ad-STE (**J**, $n = 9$) as recorded in metabolic cages. Except as otherwise indicated, data are presented as mean ± SD. NS: not significant, *$p < 0.05$ and **$p < 0.01$ as determined by unpaired two-tailed Student $t$ test (**D–F**, **I**, and **J**). The data underlying the graphs shown in the figure can be found in S1 Source Data.
(TIF)

**S2 Fig. Histological and molecular analysis of adipose tissues in SCN-lesioned and sham mice under TRF-STE and Ad-STE conditions.** (**A**) Representative H&E-stained images of eWAT collected at ZT16 under Ad-STE and TRF-STE conditions (left), with quantification of adipocyte cross-sectional areas (right). Scale bars, 50 µm. (**B**) mRNA expression of genes involved in fatty acid mobilization and oxidation in interscapular BAT from sham and SCNx mice under Ad-STE and TRF-STE ($n = 6$ per group). (**C**) Representative H&E-stained sections of interscapular BAT collected at ZT16 under Ad-STE and TRF-STE conditions. Scale bars, 50 µm. (**D**) Representative Oil Red O-stained sections of interscapular BAT collected at ZT16

under Ad-STE and TRF-STE (left), with quantification of the Oil Red O-positive area (right). Scale bars, 50 μm. Data are presented as mean±SD. *$p<0.05$, **$p<0.01$, ***$p<0.001$, and ****$p<0.0001$ as determined by unpaired two-tailed Student $t$ test (**A**, **B,** and **D**). The data underlying the graphs shown in the figure can be found in S1 Source Data.
(TIF)

**S3 Fig. Sympathetic regulation and thermogenic gene expression in BAT.** (**A**) Retrograde tracing showing terminal projections in the PVN, DMH, and ARC regions. Scale bar, 100 μm. (**B** and **C**) Western blots analysis of TH in interscapular BAT (**B**) from sham and SCNx mice at ZT4 and ZT16 under Ad-STE and TRF-STE, with densitometry analysis (**C**). $n=4$ per group. (**D** and **E**) Western blots analysis of TH (**D**) in interscapular BAT from 6-OHDA-treated mice, with densitometry analysis (**E**). $n=4$ per group. (**F** and **G**) Changes in interscapular BAT and tail temperatures in response to 6-OHDA treatment in sham and SCNx mice under Ad-STE to TRF-STE (**F**). Data presented as mean±SEM, $n=6$ per group. Representative thermographic images illustrate body surface temperature (**G**). (**H** and **I**) Changes in interscapular BAT and tail temperatures following SR59230A treatment in sham and SCNx mice under Ad-STE to TRF-STE (**H**). Data presented as mean±SEM, sham-SR mice $n=6$, SCNx-SR mice $n=5$. Representative thermographic images illustrate body surface temperature (**I**). SR: SR59230A. (**J**) SnRNA-seq analysis identified eight major cell clusters (S1 Table). (**K** and **L**) UMAP plots showing Log2 expression levels of *Pgc1α* (**K**) and *Ucp1* (**L**) in sham and SCNx mice at ZT4 and ZT16 under TRF-STE. (**M** and **N**) Mitochondrial OCR measurements using Seahorse XF Analyzer in interscapular BAT mitochondria isolated from 6-OHDA-treated sham and SCNx mice (**M**), with quantification of basal respiration, ATP-linked respiration, and maximal respiration (**N**). $n=7$ per group. Unless otherwise indicated, data are presented as mean±SD. NS: not significant, *$p<0.05$, **$p<0.01$, and ***$p<0.01$, unpaired two-tailed Student $t$ test (**C**, **E**, and **N**), two-way ANOVA with Sidak'S multiple comparisons test (**F** and **H**). The data underlying the graphs shown in the figure can be found in S1 Source Data. Raw blot images can be found in S1 Raw Images.
(TIF)

**S4 Fig. Molecular signatures of SCN-mediated BAT thermogenesis.** (**A**) A heatmap based on RNA-seq analysis of genes related to lipolysis, lipid transport, lipogenesis, and carbohydrate metabolism. (**B**) Gene expression levels involved in methylation and detoxification based on RNA-seq analysis. (fold change > 2, $p<0.01$). $n=3$ per group. (**C**) Validation of senescence markers (*Cdkn1a*, *Gadd45γ*) using qPCR in sham and SCNx interscapular BAT under Ad-STE and TRF-STE at ZT4 and ZT16. Left: RNA-seq expression values. $n=3$ per group; right: corresponding qPCR. $n=6$ per group. (**D**) Validation of proliferation-related genes (*S100b*, *Ccnd1*) by qPCR under the same conditions as in (**C**). Left: RNA-seq expression values. $n=3$ per group; right: corresponding qPCR. $n=6$ per group. (**E** and **F**) Genes significantly altered in the interscapular BAT of SCNx mice at both ZT4 and ZT16 under Ad-STE and TRF-STE (fold change > 2, $p<0.01$). $n=3$ per group. (**G** and **H**) Representative images of anti-PDGFRα immunostaining in interscapular BAT sections from sham and SCNx mice under Ad-STE and TRF-STE conditions, with 4°C and 30°C acclimation included as thermal controls (**G**). Quantification of PDGFRα-positive cells is shown in (**H**). Scale bars, 50 μm. Data are presented as mean±SD. *$p<0.05$, **$p<0.01$, ***$p<0.001$, and ***$p<0.0001$ as determined by unpaired two-tailed Student $t$ test (**B-F**, **H**). The data underlying the graphs shown in the figure can be found in S1 Source Data.
(TIF)

**S5 Fig. Analysis of *S100b* levels in brown adipocytes using snRNA-seq and ADRB3 stability assay.** (**A**) UMAP plots display Log2 expression levels of *S100b* in sham, and SCNx mice at ZT4 and ZT16 under TRF-STE. (**B**) Violin plots depicting log2 expression levels of *S100b* in adipocytes from sham and SCNx mice at ZT4 and ZT16 under TRF-STE. (**C**) Immunoblot analysis of ADRB3 stability in 293T cells overexpressing *Adrb3* and treated with recombinant S100B protein and cycloheximide (CHX). The assay was performed to evaluate the effect of S100B on ADRB3 protein stability. Raw blot images can be found in S1 Raw Images.
(TIF)

**S6 Fig. Effects of SCN manipulation on interscapular BAT thermogenesis.** (**A**) Interscapular BAT and tail temperature as indicated in responses to scramble or knockdown *S100b* in SCNx mice from Ad-STE to TRF-STE. ShRNA *S100b* mice $n = 5$, scramble RNA mice $n = 6$. Representative thermographic images illustrate body surface temperature. (**B**) Interscapular BAT and tail temperature as indicated in responses to EGFP or overexpressed *S100b* in WT mice from Ad-STE to TRF-STE, $n = 6$ per group. Representative thermographic images illustrate body surface temperature. Data presented as mean±SEM, *$p < 0.05$ and **$p < 0.01$, two-way ANOVA with Sidak'S multiple comparisons test (**A**, **B**). The data underlying the graphs shown in the figure can be found in S1 Source Data.
(TIF)

**S7 Fig. Impact of multiple SCN disruption on interscapular BAT thermogenesis.** (**A**) Locomotor screening for arrhythmic activity in mice following constant light treatment for 2 months. (**B**) Interscapular BAT and tail temperature responses in WT mice under LD or LL conditions from Ad-STE to TRF-STE. Data presented as mean±SEM, $n = 4$ per group. Representative thermographic images illustrate body surface temperature. (**C**) Schematic of NMDA treatment (left), locomotor screening (middle), and representative immunofluorescence images of GFAP staining in SCN (right). $n = 5$ per group. (**D**) Validation of AAV-DIO-Casp3-GFP and AAV-VGAT1-Cre injection in the SCN. $n = 6$ per group. (**E** and **F**) Relative mRNA expression of the indicated genes after NMDA (**E**) or Caspase-3 injecting treatment (**F**). Except as otherwise indicated, data are presented as mean±SD. *$p < 0.05$, **$p < 0.01$, and ***$p < 0.001$, unpaired two-tailed Student *t* test (**E** and **F**), two-way ANOVA with Sidak'S multiple comparisons test (**B**). Scale bars, 50 μm, (**C**) 100 μm (**D**). Schematic (**C**, **D**) created in BioRender.com. The data underlying the graphs shown in the figure can be found in S1 Source Data.
(TIF)

**S1 Table. The top 10 markers for each cluster in snRNA-seq.**
(DOCX)

**S2 Table. The cell number and frequency for each cluster in snRNA-seq.**
(DOCX)

**S3 Table. IPA of differentially expressed genes.**
(DOCX)

**S4 Table. Relative expression levels of S100b across each cluster under TRF-STE.**
(DOCX)

**S5 Table. Quantitative RT-PCR primer information.**
(DOCX)

**S1 Source Data.** Numerical values underlying all graphs in the main body and supporting information.
(XLSX)

**S1 Raw Images. Uncropped version of all Western Blot images in the main body and supporting information.**
(PDF)

## Acknowledgments

We thank members of Cam-Su GRC for their assistance in the animal facility and Xu's laboratory for discussion.

## Author contributions

**Conceptualization:** Yizhun Zeng, Antonio Vidal-Puig, Ying Xu.

**Data curation:** Yizhun Zeng, Xiaopeng Song, Qi Chen, Yue Gu.

**Formal analysis:** Yizhun Zeng, Xiaopeng Song, Qi Chen, Yue Gu, Liujia Qian, Tiannan Guo, Ying Xu.

**Funding acquisition:** Tao Wang, Ying Xu.

**Investigation:** Jie Zhang, Tao Zhou, Zhihao Li, Le Chang.

**Methodology:** Yizhun Zeng, Hongwei Yao, Yan Wang, Liyan Miao.

**Project administration:** Yizhun Zeng, Ying Xu.

**Resources:** Ying Xu.

**Supervision:** Tao Wang, Ying Xu.

**Validation:** Yizhun Zeng, Yue Gu.

**Visualization:** Yizhun Zeng.

**Writing – original draft:** Yizhun Zeng, Ying Xu.

**Writing – review & editing:** Yizhun Zeng, Xiaopeng Song, Qi Chen, Yong Zhang, Sonia Rodriguez-Fernandez, Antonio Vidal-Puig, Ying Xu.

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
