## [Editor Report · Decision Letter 0]

20 Jun 2025

Dear Dr Xu,

Thank you for submitting your manuscript entitled "Suprachiasmatic nucleus regulation of brown fat thermogenesis via ADRB3-S100B axis" for consideration as a Research Article by PLOS Biology.

Your manuscript has now been evaluated by the PLOS Biology editorial staff as well as by an academic editor with relevant expertise and I am writing to let you know that we would like to send your submission out for external peer review.

Once your full submission is complete, your paper will undergo a series of checks in preparation for peer review. After your manuscript has passed the checks it will be sent out for review. To provide the metadata for your submission, please Login to Editorial Manager (https://www.editorialmanager.com/pbiology) within two working days, i.e. by Jun 24 2025 11:59PM.

Kind regards,

Luke

Lucas Smith, Ph.D.

Senior Editor

PLOS Biology

lsmith@plos.org

---

## [Decision Letter · Decision Letter 1]

6 Aug 2025

Dear Dr Xu,

Thank you for your patience while your manuscript "Suprachiasmatic nucleus regulation of brown fat thermogenesis via ADRB3-S100B axis" was peer-reviewed at PLOS Biology. It has now been evaluated by the PLOS Biology editors, an Academic Editor with relevant expertise, and by several independent reviewers. In light of the reviews, which you will find at the end of this email, we would like to invite you to revise the work to thoroughly address the reviewers' reports.

As you will see below, the reviewers are a bit split in their assessment of your study. Reviewers 2 and 3 report that the study is well executed and their concerns should be largely addressable through textual revisions or straightforward re-analyses. Reviewer 1 is more critical of the study, noting some overlap with earlier work, questioning the physiological relevance of the model, and citing missing control data ("no comparison was made between...").

Having discussed Reviewer 1's comments with the other reviewers and the Academic Editor, we think that s/he has raised some valid points - but overall we think that the study has the potential to represent an important contribution to the field, and that his/her concerns are addressable in a revision. We think that reviewer 1's comment about the missing controls is important that this should be addressed with additional experiments if you do not have the data already. Regarding the physiological relevance of the study, we think this point could be addressed with textual changes. We encourage you to add a bit more discussion, clearly describing the methods, and that you provide further justifications and rationales for the approaches used here, in favor of their relevance. We think it would also be important for you to refine the interpretations of your study and to discuss some of the points raised by Reviewer 1 as limitations and future directions.

Given the extent of revision needed, we cannot make a decision about publication until we have seen the revised manuscript and your response to the reviewers' comments. Your revised manuscript is likely to be sent for further evaluation by all or a subset of the reviewers.

**IMPORTANT - SUBMITTING YOUR REVISION**

*Re-submission Checklist*

*Published Peer Review*

*PLOS Data Policy*

*Blot and Gel Data Policy*

Sincerely,

Luke

Lucas Smith, Ph.D.

Senior Editor

PLOS Biology

lsmith@plos.org

REVIEWS:

Reviewer #1: Zheng et al present a study whereby mice were kept at sub thermal conditions, changed into a new cage without food at ZT 20 (four hours before light on) kept without food until the next ZT 16, then given food till ZT 20. Intact (Sham lesioned) animals exposed to these conditions were compared with those having a Suprachiasmatic Nucleus (SCN) lesion. No comparison was made between intact animals in normal conditions and the food-restricted groups to show the impact of this food restriction on the normal physiology.

Next, the authors demonstrate that the SCN influences brown adipose tissue (BAT) through the sympathetic nervous system. Followed by an analysis of the involved genes in BAT using SCN-intact and lesioned animals. Hereafter, the study focuses on the S100B calcium-binding protein.

A 48-hour fast completely suppressed S100b, whereas high-fat diet (HFD) feeding significantly increased its expression. Interestingly thermoneutral acclimation (30 °C) markedly suppressed S100b, while cold exposure (4 °C) did not influence S100b. These findings demonstrate that S100b is especially regulated by nutrient availability rather than cold stress. This seems in contrast with the conclusion of the paper that the SCN influences BAT thermogenesis via S100B signaling. The reason for discrepancy might be that the authors only use SCN lesioning or by disrupting SCN functioning in other ways. The observation that SCN lesioning promotes S100B activation might be just related with the fact that SCN lesioning is known to prevent the sleep associated temperature decrease.

Critique: For this reviewer it was very difficult to understand the rationale of the present study. Since the study of Liu et al (ref 12) it is known that lesioning the SCN results in the prevention of the sleep associated temperature dip, SCN lesioning keeps the temperature high. This means that the SCN does not stimulate so much the sympathetic output to BAT but rather inhibits it, resulting in a dip in body temperature in the beginning of the sleep phase. Consequently, also the fasting induced temperature dip in the beginning of the sleep phase does not occur after SCN lesioning. Furthermore, it has already been known for long that the SCN can influence BAT physiology by influencing the sympathetic innervation of BAT(Bamshad et al A.J.Phys. 1999).

Furthermore, the imposed acute(!) change in food availability, as used in the present paper, influences many SCN associated systems that normally regulate the body temperature depending on the availability of food (see e.g. a review of Mendez-Hernandez et al 2020 Obesity).

Suggestion: The authors indicate several possibilities to promote an interest in protein S100B in association with the functioning of the SCN. I would suggest that the authors use a more straightforward model in which the known functions of the SCN: deepening the temperature trough during fasting at ZT 0-2 is associated with an increase in temperature to normal high levels during activity. This approach would have the possibility to reveal the functioning of S100B in association with the functioning of the SCN. Now the authors use a model (SCN-lesioning) that was very useful in 2002 (ref12) but since then it was shown that the SCN influences sympathetically BAT (Bamshad), manages in interaction with the arcuate nucleus the daily temperature increase and decrease (Guzman-Ruiz J. Neurosci. 2015). When the authors want to clarify a role for S100B in the temperature regulation of the SCN and brown adipose tissue metabolism it would be more logical to focus on the normal daily temperature changes associated with the SCN driven anticipation of activity and rest that are more pronounced during fasting conditions (REF 12).

Reviewer #2: In this study, the authors investigate the role of the suprachiasmatic nucleus (SCN) in the regulation of brown adipose tissue (BAT) thermogenesis under conditions of time-restricted feeding in subthermoneutral environments (TRF-STE). They show that SCN lesioning preserves BAT thermogenic capacity while disrupting substrate using, specifically by impairing lipid mobilization and promoting glucose-driven thermogenesis. These effects are mediated through alterations in sympathetic nervous system signaling, indicating that the SCN modulates thermogenesis via this way. They also demonstrate that SCN lesioning reverses TRF-STE-induced cellular senescence in BAT and promotes cell proliferation through the S100B, a protein identified as a nutrient-sensitive effector. S100B amplifies β3-adrenergic signaling and enhances thermogenic capacity. The study further demonstrates that S100B is both necessary and sufficient to mediate SCN-dependent thermogenic resilience in BAT, highlighting its central role in the metabolic adaptation of this tissue. The authors show that S100B physically interacts with the β3-adrenergic receptor (ADRB3), modulating its abundance and potentially enhancing its signaling responsiveness. Overall, their results identify the ADRB3-S100B axis as a key adaptive mechanism in BAT following SCN disruption.

The work is of high quality, as is the writing of the manuscript and the presentation of the data. The amount of work is impressive, with a wide variety of models and methods to support the conclusions. Here are some minor comments that are nonetheless worth addressing:

- In general, for most experiments, the authors should indicate in the text or figure legend how long after SCN ablation the experiments were performed. For example, changes in body composition take time and the data in Figure 1 G-I should be supplemented with the time taken to obtain such changes after SCN ablation. How do the authors explain that actograms show complete desynchronization of the circadian clock, while EE and temperature still show a circadian profile in SCNx animals? Does food intake also retain a circadian profile?

- Fig 2: The increase in lipid droplet size in SCNx mice is not evident, particularly in Ad-STE. The authors should perform quantification to support their conclusion of larger lipid droplets in SCN-injured MTDs in both Ad-STE and TRF- STE. Oil-Red-O staining would also be informative.

- Fig 3: the authors should indicate the number of samples used in this experiment and quantify the signals in each animal. To conclude that SCN modulates thermogenic capacity via sympathetic signaling, OCR should be measured in mitochondria isolated from the interscapular BAT of sham-6-OHDA and SCNx-6-OHDA mice, and in sham and SCNx mice treated with the ADRB3 antagonist.

- The authors should clarify whether S100B is expressed and modified in other cell types, based on the snRNAseq shown in Fig3J. What is the contribution of the other cell types? Similarly, to assert that EdU staining demonstrates preadipocyte proliferation in vivo, the authors should use a preadipocyte-specific marker in addition to EdU staining.

- The authors should discuss the relevance of changes in adipocyte proliferation in this context, in relation to the use of energy substrates.

- The graphical abstract need to be improved, in particular by adding TRF and ZT.

Reviewer #3: Zeng et al. provide a well-written and well-presented series of studies which support the conclusion that the SCN affects BAT themogenic plasticity via the SNS and ADRB3-S100B axis.

I have no major comments, but several minor ones. I present the minor comments mostly in order in which they appear in the manuscript.

Results.

Section 1, paragraph 2. "To account for these compositional differences…" This section is written as if the variable of fat or lean mass is what is compared to EE. However, the graph indicates that it is not body composition (fat or lean), but whole body weight that is tested against EE. This analysis allows one to conclude that SCNx reduces EE relative to body weight. Given the increase in body weight variability in the SCNx group (compared to WT), this test is appropriate. However, the text description of why the test was done (to look at composition differences) does not appear accurate. The final sentence of this paragraph should be edited to capture this nuance.

Discussion section. Related to the comment above, the discussion indicates that GLM accounted for differences in body composition. While that may be true, the graph in the figure suggestions otherwise.

Section 1, paragraph 3. "HFD-fed mice still experienced significant hypothermia". This description is a bit too vague for a results section - especially because in the 1st paragraph of this section the authors do a nice job of describing the BAT, core, and tail temperature. The sentence describing Fig 1L should be edited so that the reader is aware that this is only referring to core temperature. This edit doesn't change the authors' conclusions, merely provides a more accurate and full description of the data.

Section 3, paragraph 2. "…abolished the elevated thermogenic response observed in SCN lesioned mice…". While I agree with the authors that this is the likely interpretation, the graph/results are not comparing "SCNx with 6-OHDA" to "SCNx without 6-OHDA". The results would be more accurately stated as "6-OHDA treatment produces thermogenic responses that are indistinguishable between sham and SCN lesioned animals" (or similar). This edit doesn't change the authors' conclusions, merely provides a more accurate presentation of the data.

Section 3, paragraph 2. Similar comment as above, but now in reference to the ADRB3 antagonist.

Section 4, first sentence. "molecular" likely is meant to be "mechanism" or "molecular mechanism"

Section 4, second to last paragraph, last sentence. "…indicating that SCN activity modulates BAT…". The word "indicating" is too strong for the experiments described since the authors have not yet presented the data that would best support the role of the SCN yet. I would advise changing this word to "suggesting" or "supporting the idea".

Section 5, third paragraph. "This intervention ____ the expression levels of S100b…" The authors are missing a word between "intervention" and "the expression levels".

Section 6. Please describe Fig 6V in the results section and/or provide a narrative in the discussion.

Discussion Section.

The authors note in Results, section 5, that the present results contrast with prior publications which find increased serum S100B with obesity and neuroinflammation. Can the authors comment on the possible discrepancy between prior studies and theirs?

Other.

There is no mention as to whether the mice used were males, females, or a mix of each. Please clearly indicate the sex used at each mention of mice in the method section (e.g., whenever the age of the mice is mentioned, so should the sex). If using of a male-only study design, justify the rationale and add "in male mice" to paper title.

---

## [Decision Letter · Decision Letter 2]

6 Nov 2025

Dear Dr Xu,

Thank you for your patience while we considered your revised manuscript "Suprachiasmatic nucleus regulation of brown fat thermogenesis via  ADRB3-S100B axis in male mice" for publication as a Research Article at PLOS Biology. This revised version of your manuscript has been evaluated by the PLOS Biology editors, the Academic Editor and the original reviewers.

As you will see, both reviewers 1 and 3 are fully satisfied by the revision, and reviewer 2 agrees the study has been strengthened. However, reviewer 2 has flagged that the control animals shown in Figure 1 appear to have originated from independent experiments and we think that this would need to be made much clearer. If the new data was generated separately, this should be explicitly stated in the text and figure legend and supported by an appropriate reference (if the data is from a previous study). Additionally, we think you should represent the corresponding curve as a dashed line in the Figure.

Based on the reviews, we are likely to accept this manuscript for publication, provided you satisfactorily address this remaining point.

**IMPORTANT: As you revise your study we ask that you please also make sure to address the following data and other policy-related requests.

1) TITLE: We would like to suggest a tweak to your title. If you agree, we suggest you change it to:

"The suprachiasmatic nucleus regulates brown fat thermogenesis in male mice through an adrenergic receptor ADRB3-S100B signaling pathway"

^we are happy to discuss this further if you prefer a different title.

2) ETHICS STATEMENT: Please update your methods section to include the protocol/license number for the protocol approved by the Animal Care and Use Committee of Soochow University. Please also include the specific national or international regulations/guidelines to which your animal care and use protocol adhered. Please note that institutional or accreditation organization guidelines (such as AAALAC) do not meet this requirement.

3) DATA: Thank you for providing the underlying data for your study, which for the most part looks good. However, I was not able to access the data you provided on GEO. Can you provide me with a reviewer token so I can check that it meets our requirements? (Sorry if I missed that somewhere).

4) CODE: Per journal policy, if you have generated any custom code during the course of this investigation, please make it available without restrictions. Please ensure that the code is sufficiently well documented and reusable, and that your Data Statement in the Editorial Manager submission system accurately describes where your code can be found.

We expect to receive your revised manuscript within two weeks.

*Published Peer Review History*

*Press*

Sincerely,

Lucas

Lucas Smith, Ph.D.

Senior Editor

lsmith@plos.org

PLOS Biology

Reviewer remarks:

Reviewer #1: I consider the manuscript with the current changes as acceptable.

Reviewer #2: The quality of the main figures in this PDF version was suboptimal, most likely due to a formatting issue during the upload process. Nevertheless, the authors have conducted substantial additional work that considerably enhances the overall quality of the manuscript.

In response to my previous comments, the authors have provided the requested information in the figure legends, quantified adipocyte size, performed additional mitochondrial respiration experiments, carried out complementary analyses on the sn-RNAseq data, and added new immunostaining experiments that further support the proliferative nature of preadipocytes. They have also addressed the discussion points raised in the previous round of review and modified the graphical abstract accordingly. Taken together, these revisions and additions have greatly improved the manuscript, and I thank the authors for their efforts and clarifications.

With regard to the comments made by the other reviewers, particularly those concerning the control animals shown in Figure 1, I wonder whether these animals originate from independent experiments. If that is the case, this information should be explicitly mentioned in the text with an appropriate reference, and the corresponding curve should be represented as a dashed line in the figure to indicate this distinction.

Overall, the authors' responses are well thought out and have significantly strengthened the manuscript through the inclusion of new controls, additional experiments, and an improved discussion section. It is also important to emphasize the broad scope of this study, which includes an impressive range of models, experiments, and results. For all these reasons, I recommend the manuscript for publication.

Reviewer #3: The author addressed all my comments.

---

## [Editor Report · Decision Letter 3]

13 Nov 2025

Dear Dr Xu,

Thank you for the submission of your revised Research Article "The suprachiasmatic nucleus regulates brown fat thermogenesis in male mice through an adrenergic receptor ADRB3-S100B signaling pathway" for publication in PLOS Biology, and thank you for addressing the last reviewer and editorial requests in this revision. On behalf of my colleagues and the Academic Editor, Sebastien G Bouret, I am pleased to say that we can in principle accept your manuscript for publication, provided you address any remaining formatting and reporting issues. These will be detailed in an email you should receive within 2-3 business days from our colleagues in the journal operations team; no action is required from you until then. Please note that we will not be able to formally accept your manuscript and schedule it for publication until you have completed any requested changes.

PRESS

Sincerely, 

Lucas Smith, Ph.D.

Senior Editor

PLOS Biology

lsmith@plos.org